# Cytoneme delivery of Sonic Hedgehog from ligand-producing cells requires Myosin 10 and a Dispatched-BOC/CDON co-receptor complex

Eric T Hall[1], Miriam E Dillard[1], Daniel P Stewart[1], Yan Zhang[1], Ben Wagner[2], Rachel M Levine[1,3], Shondra M Pruett-Miller[1,3], April Sykes[4], Jamshid Temirov[1], Richard E Cheney[5], Motomi Mori[4], Camenzind G Robinson[2], Stacey K Ogden[1]*

[1]Department of Cell and Molecular Biology, St. Jude Children's Research Hospital, Memphis, United States; [2]Cell and Tissue Imaging Center, St. Jude Children's Research Hospital, Memphis, United States; [3]Center for Advanced Genome Engineering, St. Jude Children's Research Hospital, Memphis, United States; [4]Department of Biostatistics, St. Jude Children's Research Hospital, Memphis, United States; [5]Department of Cell Biology and Physiology, University of North Carolina School of Medicine, Chapel Hill, United States

**Abstract** Morphogens function in concentration-dependent manners to instruct cell fate during tissue patterning. The cytoneme morphogen transport model posits that specialized filopodia extend between morphogen-sending and responding cells to ensure that appropriate signaling thresholds are achieved. How morphogens are transported along and deployed from cytonemes, how quickly a cytoneme-delivered, receptor-dependent signal is initiated, and whether these processes are conserved across phyla are not known. Herein, we reveal that the actin motor Myosin 10 promotes vesicular transport of Sonic Hedgehog (SHH) morphogen in mouse cell cytonemes, and that SHH morphogen gradient organization is altered in neural tubes of *Myo10*$^{-/-}$ mice. We demonstrate that cytoneme-mediated deposition of SHH onto receiving cells induces a rapid, receptor-dependent signal response that occurs within seconds of ligand delivery. This activity is dependent upon a novel Dispatched (DISP)-BOC/CDON co-receptor complex that functions in ligand-producing cells to promote cytoneme occurrence and facilitate ligand delivery for signal activation.

*For correspondence:
stacey.ogden@stjude.org

## Introduction

The Hedgehog (HH) pathway is an evolutionarily conserved signaling relay that contributes to embryonic development through influencing cell fate, cell proliferation, cell and tissue polarity, stem cell maintenance, and tissue homeostasis (reviewed in *Briscoe and Thérond, 2013*). During development, HH family morphogens, which include HH in flies and Sonic (SHH), Desert (DHH) and Indian (IHH) Hedgehogs in vertebrates, function in concentration-dependent manners to instruct tissue morphogenesis through Patched (PTCH) receptors and Cell Adhesion Molecule-Related/Down-Regulated by Oncogenes (CDON), Brother of CDON (BOC) or Growth Arrest-Specific 1 (GAS1) co-receptors (*Allen et al., 2007*; *Marigo et al., 1996*; *Okada et al., 2006*; *Yao et al., 2006*). Binding of SHH to PTCH receptor complexes allows for activation of the G-protein-coupled receptor Smoothened (SMO), which can signal through both G-protein-dependent and independent effector routes to induce downstream responses (reviewed in *Arensdorf et al., 2016*). G-protein-dependent noncanonical SMO signals induce transcription-independent responses including cytoskeletal

**eLife digest** During development, cells must work together and talk to each other to build the organs and tissues of the growing embryo. To communicate precisely with long-distance targets, cells can project a series of thin finger-like structures known as cytonemes. Cells use these miniature highways to exchange cargo and signals, such as the protein sonic hedgehog (SHH for short). Alterations to the way SHH is exchanged during development predispose to cancer and lead to disorders of the nervous system.

Yet, the mechanisms by which cytonemes work in mammals remain to be fully elucidated. In particular, it is still unclear how the structures start to form, and how the proteins are loaded and transported from one end to another. A 'molecular motor' called myosin 10, which can carry cargo along the internal skeleton of cells, may be involved in these processes.

To find out, Hall et al. used fluorescent probes to track both myosin 10 and SHH in mouse cells, showing that myosin 10 carries SHH from the core of the signal-producing cell to the tips of cytonemes. There, the protein is passed to the target cell upon contact, triggering a quick response.

SHH also appeared to be more than just passive cargo, interacting with another group of proteins in the signal-emitting cell before reaching its target. This mechanism then encourages the signalling cells to produce more cytonemes towards their neighbours.

SHH is crucial during development, but also after birth: in fact, changes to SHH transport in adulthood can also disrupt tissue balance and hinder healing. Understanding how healthy tissues send this signal may reveal why and how disease emerges.

rearrangement for cell migration, lipid metabolic responses, and $Ca^{2+}$ release (*Arensdorf et al., 2017*; *Belgacem and Borodinsky, 2011*; *Ho Wei et al., 2018*). The canonical SMO effector route is thought to be largely G-protein-independent, and requires SMO translocation into the primary cilium for activation of GLI transcriptional effectors to induce target gene expression (*Arensdorf et al., 2016*; *Corbit et al., 2005*).

To induce appropriate responses in target cells, HH ligands must deploy from their site of synthesis and transport to short- and long-range target cells. HH family ligands are unique in that they are lipid-modified by a covalently linked cholesterol moiety on their carboxyl termini, and by a long-chain fatty acid such as palmitate on their amino termini (*Pepinsky et al., 1998*; *Porter et al., 1996*). These modifications result in high affinity of the ligand for producing-cell membranes, necessitating specific deployment and transport mechanisms to assure the formation and fitness of HH morphogen gradients during tissue patterning (reviewed in *Hall et al., 2019*). In vertebrates, proteins dedicated to deployment of SHH include the twelve-transmembrane spanning protein Dispatched (DISP) and the secreted glycoprotein SCUBE2 (*Burke et al., 1999*; *Creanga et al., 2012*; *Tukachinsky et al., 2012*). Mechanisms by which DISP and SCUBE2 collaborate to ensure SHH deployment and influence its establishment of its morphogen gradient are not yet clear, but several models have been proposed.

The cytoneme model posits that long, specialized filopodia extend from producing and receiving cells to facilitate transport and exchange of HHs across developing tissues (*Kornberg and Roy, 2014*; *Ramírez-Weber and Kornberg, 1999*). Cytonemes are thin filopodia, typically smaller than 200 nm in diameter, that can reach up to ~300 μm from the originating cell body (*Kornberg, 2014*). Cytonemes functioning during morphogenesis were first recognized in *Drosophila* wing imaginal discs (*Ramírez-Weber and Kornberg, 1999*). Subsequent studies found them to contain signaling molecules including HH, WNT, TGFβ, Notch ligands, and growth factors (reviewed in *Fairchild and Barna, 2014*; *Kornberg, 2014*). Interrogations of HH-containing cytonemes in fly and chick systems revealed localization of DISP, BOC (BOI in *Drosophila*), CDON (iHOG, *Drosophila*), and PTCH to the cellular structures along with ligand (*Bischoff et al., 2013*; *Bodeen et al., 2017*; *Chen et al., 2017*; *Gradilla et al., 2014*; *Sanders et al., 2013*). The cytoneme-specific functionalities of these proteins, and how they facilitate HH ligand movement, have not yet been determined. Although specialized filopodia containing SHH have been observed in vertebrate systems (*Sanders et al., 2013*), little is known about their cell biology in mammals. The full cast of proteins dedicated to initiation, growth,

and maintenance of cytonemes are not yet known, nor are the mechanisms by which morphogens are loaded into and transported along the cellular structures.

Herein we interrogate cytoneme-based transport of SHH in mouse cells. By using newly developed imaging protocols and developing a rapid read-out for SHH pathway induction, we reveal that Myosin 10 (MYO10) is required for movement of SHH to cytoneme tips to initiate a signal response in target cells. We find that disruption of MYO10 in vivo leads to neural tube patterning defects consistent with attenuated SHH morphogen gradient function, confirming that cell-based interrogation of cytonemes can predict in vivo relevance. Our studies also reveal that a novel complex between SHH, DISP, and BOC/CDON contributes to SHH cytoneme occurrence and ligand delivery, and that BOC/CDON activity in ligand-producing cells is required for cytoneme-mediated induction of an SHH signal response in target cells.

## Results

### SHH promotes cytoneme formation

To interrogate cytoneme-based SHH transport in mammalian systems, NIH3T3 cells and mouse embryonic fibroblasts (MEFs) expressing SHH plus the membrane marker mCherry-Mem (mCherry fused to the first 20 residues of neuromodulin) were fixed using the MEM-fix technique, and then examined by confocal microscopy (*Figure 1A–B''*; *Bodeen et al., 2017*; *Hall and Ogden, 2018*). Image analysis of the mCherry-Mem signal in SHH-expressing cells revealed long extensions from both NIH3T3 cells and MEFs that reached around and over neighboring cells (*Figure 1A–B''*). Depth analysis of the mCherry-Mem signal revealed that cytoneme-like projections originated from portions of the cell membranes that were not in contact with the growth surface (*Figure 1A'–A''', B''*). The small diameter, long lengths, and growth patterns of these extensions are consistent with the documented characteristics of cytonemes, indicating NIH3T3 cells and MEFs can be used to interrogate the specialized filopodia (*Bodeen et al., 2017*; *Hall and Ogden, 2018*; *Kornberg and Roy, 2014*; *Ramírez-Weber and Kornberg, 1999*).

The atypical actin-based motor protein MYO10 is thought to promote filopodial outgrowth through facilitating anterograde transport of protein cargo that supports growth and maintenance of the cellular outgrowths (*Bohil et al., 2006*; *Hirano et al., 2011*; *Tokuo and Ikebe, 2004*; *Wei et al., 2011*; *Zhang et al., 2004*). We tested for localization of MYO10 to SHH-containing projections by expressing GFP tagged MYO10 in NIH3T3 cells. Indeed, MYO10-GFP was enriched in tips of projections from SHH-expressing cells, and showed co-localization with SHH puncta in a subset of the extensions (*Figure 1C,C'*). Live imaging revealed that the MYO10-tip-enriched extensions were highly dynamic and capable of forming stable connections with MYO10-positive extensions from neighboring cells (*Videos 1–3*). Shorter extensions that maintained contact with the growth substrate were largely immobile, suggesting they were likely adhesion or retraction fibers, rather than dynamic cytonemes (*Videos 1* and *2*; *Kornberg and Roy, 2014*; *Zhang et al., 2004*).

Cytoneme-like structures were detected in the absence of SHH expression in both MEFs and NIH3T3 cells, indicating that the morphogen is not required for their initiation. However, SHH expression raised the cytoneme occurrence rate in both cell types from ~30% to ~60% (*Figure 1D*), and more than doubled the average number of cytonemes per NIH3T3 cell (*Figure 1E*). To test whether increased cytoneme occurrence rates correlated with SHH expression level, cytonemes were quantified in HEK cells stably transfected with a ponasterone A-inducible SHH expression vector (*Goetz et al., 2006*). Likely due to a low level of 'leaky' SHH expression occurring in the absence of induction, baseline cytoneme occurrence in inducible cells was increased compared to untransfected controls (*Figure 1—figure supplement 1A–F*). Addition of increasing concentrations of ponasterone A led to a dose-dependent increase in cytoneme occurrence that correlated with increased SHH protein production (*Figure 1—figure supplement 1B–F*). These results suggest SHH-producing cells may tune cytoneme occurrence rates proportional to morphogen expression levels.

To test whether the SHH signal transducing protein SMO contributed to increased cytoneme occurrence rates observed in cultured cells expressing ligand, NIH3T3 cells were treated with the direct SMO agonist SAG and antagonist vismodegib (*Figure 1—figure supplement 1G*). SAG failed to significantly alter cytoneme occurrence in the absence or presence of SHH, indicating that direct induction of SMO is not sufficient to induce a cytoneme response. Vismodegib treatment raised

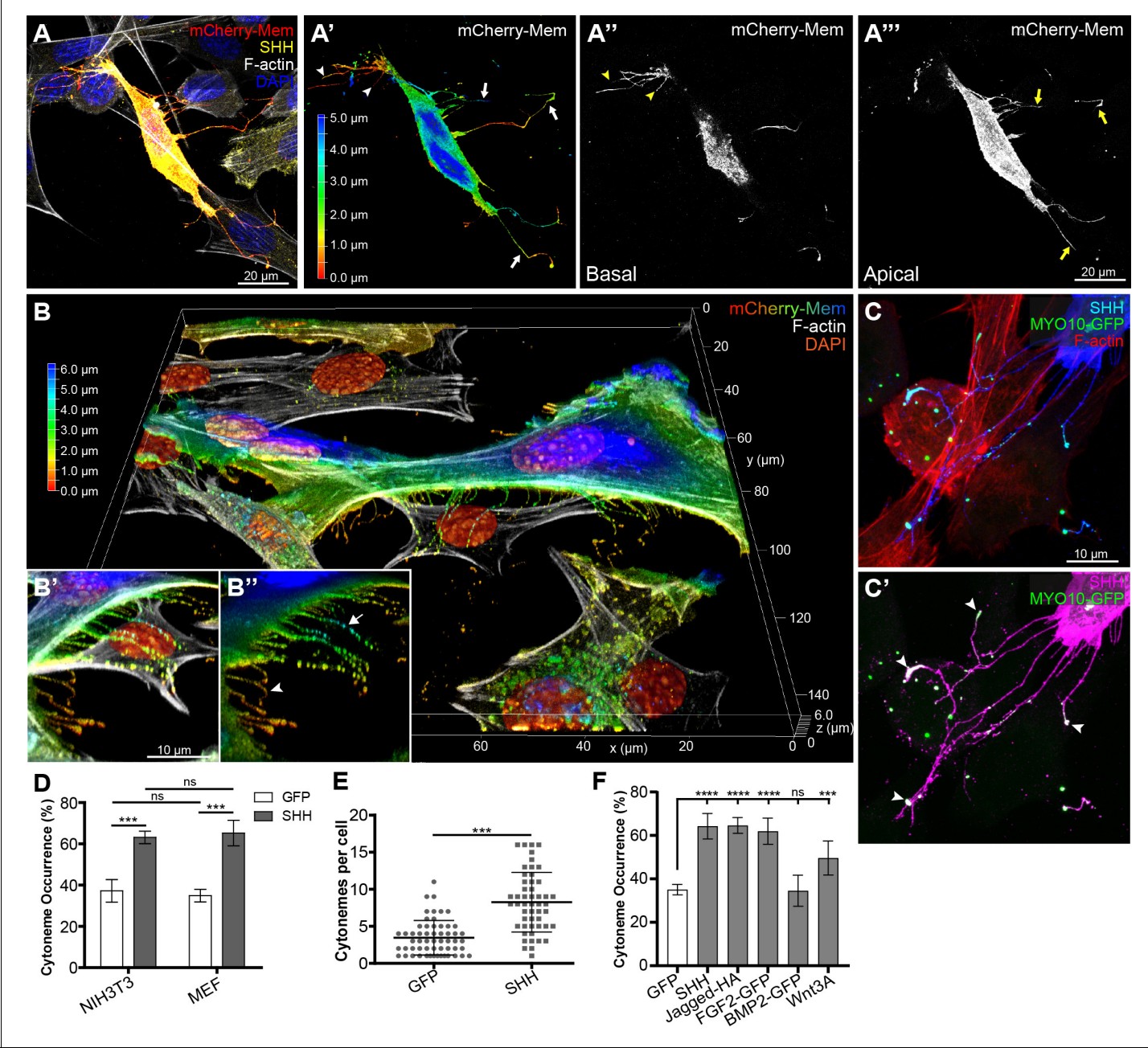

**Figure 1.** SHH promotes cytoneme occurrence. (A–A''') Cytonemes of a SHH (yellow) and mCherry-Mem (red) expressing NIH3T3 cell contacting neighboring cells is shown. (A') mCherry-Mem signal represented as a Z-axis depth colored projection showing filopodia (arrowheads) and cytonemes (arrows). (A'') An image of cell membrane protrusions that maintain contact with the culture coverslip shows that coverslip-adjacent projections are shorter (arrowheads). (A''') A projection of the cell membrane not in contact with the coverslip shows cytonemes (arrows). (B–B'') A 3D render of an MEF cell expressing SHH and mCherry-Mem colored for Z-axis depth shows cytonemes reaching around a neighboring cell (zoom in and rotation B',B''). Coverslip-adjacent projecting filopodia are orange (arrowhead), with a cytoneme example indicated by an arrow in B'' (green). (C) An NIH3T3 cell expressing MYO10-GFP and SHH (blue). F-actin is shown in red. SHH signal is saturated in C' (magenta) to visualize cytonemes. MYO10-GFP accumulates in puncta at cytoneme tips with SHH (C' arrowheads). (D) Cytoneme occurrence rates were calculated in MEM-fixed NIH3T3 and MEF cells in the absence and presence of SHH. (E) The number of cytonemes per NIH3T3 cell were counted in the presence of either GFP (n = 57) or SHH expression (n = 51). (F) Cytoneme occurrence rates in NIH3T3 cells expressing GFP, SHH, Jagged-HA, FGF2-GFP, BMP2-GFP, or Wnt3A were quantified. Data are represented as mean ± SD. ns = not significant, ***p<0.001, ****p<0.0001.

The online version of this article includes the following figure supplement(s) for figure 1:

**Figure supplement 1.** SHH promotes cytoneme formation in a concentration dependent manner.

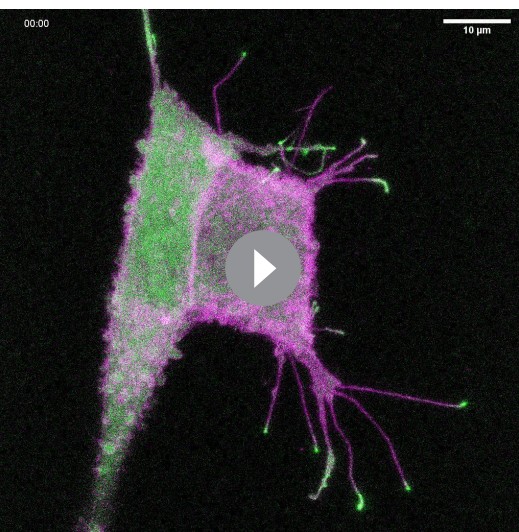

**Video 1.** Cytonemes are dynamic cellular extensions that contain MYO10. Two adjacent NIH3T3 cells expressing SHH, mCherry-Mem (magenta), and MYO10-EGFP (green) are shown as a maximum intensity projection of 10 Z-sections spanning 4.5 μm, imaged at 4.1 s/frame over 18 min. Cytonemes are distinguished from other cellular extension by active growth and accumulation of MYO10-EGFP at the tips. Time stamp indicates min:s.

https://elifesciences.org/articles/61432#video1

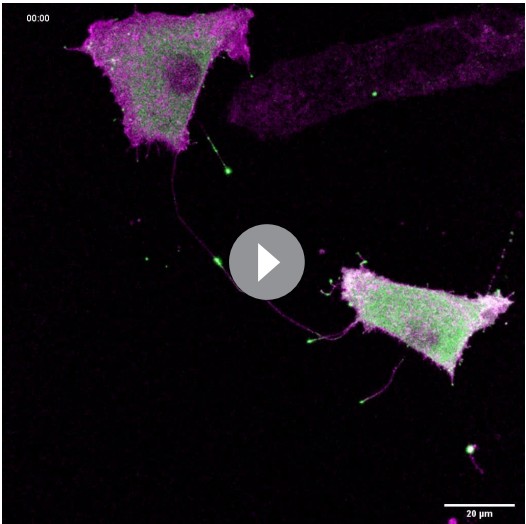

**Video 3.** Cytonemes exhibit transient interactions and stable connections. NIH3T3 cells expressing SHH, mCherry-Mem (magenta), and MYO10-EGFP (green) are shown as a maximum intensity projection of 5 Z-sections spanning 2 μm, imaged at 2.05 s/frame over 9 min. Cytonemes transiently scan membrane of a neighboring cell (upper right) and form stable connections with cytonemes from adjacent cells (center). MYO10-GFP moves along cytonemes in puncta and enriches at cytoneme contact points. Time stamp indicates min:s.

https://elifesciences.org/articles/61432#video3

baseline cytoneme occurrence and blunted the ability of SHH to increase occurrence rate over the elevated baseline. To clarify whether these changes resulted from loss of SMO induction, cytonemes were quantified in CRISPR generated $Smo^{-/-}$ NIH3T3 cells. $Smo^{-/-}$ cells showed an ~15% increase in occurrence rates in response to SHH expression, indicating that SMO is not required for a cytoneme response. However, the increase in occurrence rate was reduced compared to what was observed in control cells (~28% increase), suggesting active SMO may enhance the ability of SHH to increase cytonemes in cultured cells.

The ability of SHH to induce cytonemes independent of its canonical signal transducing protein SMO suggests the morphogen may act in a cell autonomous manner to control its own deployment. To determine whether other morphogens or developmental signaling molecules documented to localize to cytonemes might also be able to influence cytonemes, we quantified occurrence rates in NIH3T3 cells expressing the Notch ligand Jagged, BMP2, FGF2, or Wnt3A (*Figure 1F*). Like SHH, Jagged and FGF2 triggered an approximate doubling of cytoneme occurrence. Wnt3A also increased occurrence rates, albeit to a lesser extent. BMP2 expression did not alter baseline cytoneme occurrence, indicating that some, but not all developmentally relevant signaling proteins can

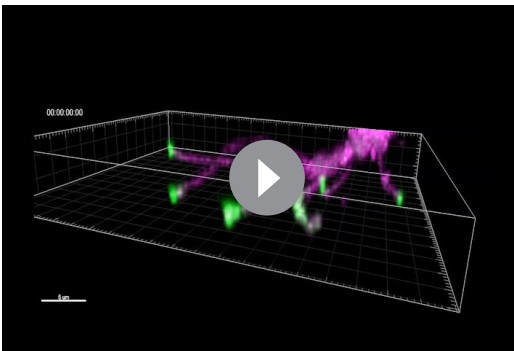

**Video 2.** Dynamic cytonemes move in three dimensions. Lateral projection of a cell edge from *Video 1*, imaged as described in *Video 1*. Active cytoneme extensions frequently traverse through the media, eventually dropping onto the culture surface. Time stamp min:s.

https://elifesciences.org/articles/61432#video2

modulate activity of the specialized filopodia in NIH3T3 cells.

## Cytonemes deliver an SHH activation signal

To determine whether SHH-containing cytonemes transmitted an activation signal to receiving cells, we developed a contact-mediated activation assay in which SHH pathway induction in receiving cells could be rapidly detected. The current temporal indicator of pathway induction tracks accumulation of the SHH signal transducing G-protein-coupled receptor Smoothened (SMO) into primary cilia (*Corbit et al., 2005*). However, both active and inactive SMO proteins cycle through the primary cilium, making this assay sub-optimal for tracking SMO activation in real time (*Milenkovic et al., 2015*; *Rohatgi et al., 2009*). Active SMO signals through Gαi heterotrimeric G proteins to raise intracellular $Ca^{2+}$, which we reasoned would be a rapid and activation-specific read-out for pathway induction resulting from cytoneme-based ligand delivery (*Adachi et al., 2019*; *Belgacem and Borodinsky, 2011*; *Klatt Shaw et al., 2018*). A previous study examining filopodial-based transport of SHH used a truncated SHH-N construct, which is amenable to palmitoylation, but lacks the carboxyl-terminal cholesterol modification (*Sanders et al., 2013*). Because cholesterol contributes to both SHH morphogen gradient formation and PTCH binding on receiving cells (*Gong et al., 2018*; *Li et al., 2006*), we wanted to ensure that we analyzed trafficking of the physiologically relevant dually lipidated molecule. We generated SHH-GFP/mCherry such that both lipid modifications are added to a signaling molecule with an internal GFP or mCherry tag (*Figure 2—figure supplement 1A–B'*). Transcriptional reporter assays confirmed functionality of the internally-tagged fluorescent SHH proteins (*Figure 2—figure supplement 1C,C'*). Once SHH expression constructs were validated, we monitored $Ca^{2+}$ flux by co-culturing R-GECO-expressing $Ca^{2+}$ reporter cells with NIH3T3 cells expressing palmitoylated and cholesterol-modified SHH-GFP or GFP control. R-GECO cells in contact with cytonemes, but not cell body, from control or SHH GFP-positive cells were monitored for $Ca^{2+}$ reporter flux (*Figure 2A–B'* and *Videos 4–5*). To increase the likelihood that signals would result from deposition of SHH via cytonemes, and not from SHH secreted into the media, media wash-out was performed at 15 min intervals for the duration of data acquisition. We counted any R-GECO $Ca^{2+}$ flux with a minimum peak fluorescence of 50 to be a positive event (*Figure 2B–C'*). This flux value was determined by examining the distribution of fluorescence intensity of R-GECO cells in contact with GFP control cells (n = 27 cells). A value of 50 was determined to be within the 90th and 95th percentile of flux value distributions in control cells (*Figure 2B*), indicating any flux over 50 would likely be significantly above the control intensity range. Consistent with a SHH-induced signal, $Ca^{2+}$ flux rates of R-GECO reporter cells in contact with SHH containing cytonemes were more than twice the rate of flux observed in reporter cells contacted by cytonemes from GFP-expressing control cells (*Figure 2B,B', D*). Furthermore, R-GECO cells in continuous contact with SHH-containing cytonemes had a significantly higher total time spent in flux than cells in contact with control cell cytonemes (*Figure 2E*). R-GECO reporter cells typically produced transient $Ca^{2+}$ pulses within ~10–20 s of SHH-GFP release from cytonemes docked to their cell membranes (*Figure 2B'*, red dashes). To determine whether there was a statistically significant correlation between cytoneme-mediated ligand delivery and $Ca^{2+}$ response, we documented all flux events with a mean intensity of over 50 that occurred within 20 s of SHH deposition (n = 15 cells). A Wilcoxon signed rank test performed against these results confirmed that reporter cells spent a significantly greater amount of time in positive flux within a 20 s window following an SHH deposit than they did outside this response window (p=6.1e-05, *Figure 2F*). As such, a significant correlation between SHH delivery and $Ca^{2+}$ release was confirmed. Importantly, SHH-stimulated $Ca^{2+}$ flux was blocked in $Smo^{-/-}$ R-GECO cells (*Figure 2D* columns 5 and 6) or by treatment with the inverse SMO agonist cyclopamine (*Figure 2C–D* columns 7 and 8), confirming specificity of the $Ca^{2+}$ response to SHH pathway activation. To confirm that signal induction resulted from SHH delivered through cytonemes, and not from SHH ligand that may have been secreted into the culture medium from ligand-expressing cells, GFP and SHH-GFP expressing cells were co-cultured with R-GECO reporter cells. Flux was monitored in reporter cells in contact with cytonemes from SHH-GFP or GFP-expressing control cells (*Figure 2D*). Reporter cells in contact with SHH-GFP-expressing cytonemes maintained higher flux rates than reporter cells in contact with GFP-expressing control cells, and exhibited a flux rate similar to what was observed in SHH-GFP/R-GECO culture conditions (*Figure 2D* column 2 vs. column 4). Reporter cells in contact with GFP-expressing cells in the mixed culture exhibited a flux rate similar to reporter cells cultured with GFP-expressing cells in the absence of SHH-GFP co-culture (column 1 vs. column 3). We conclude secreted SHH

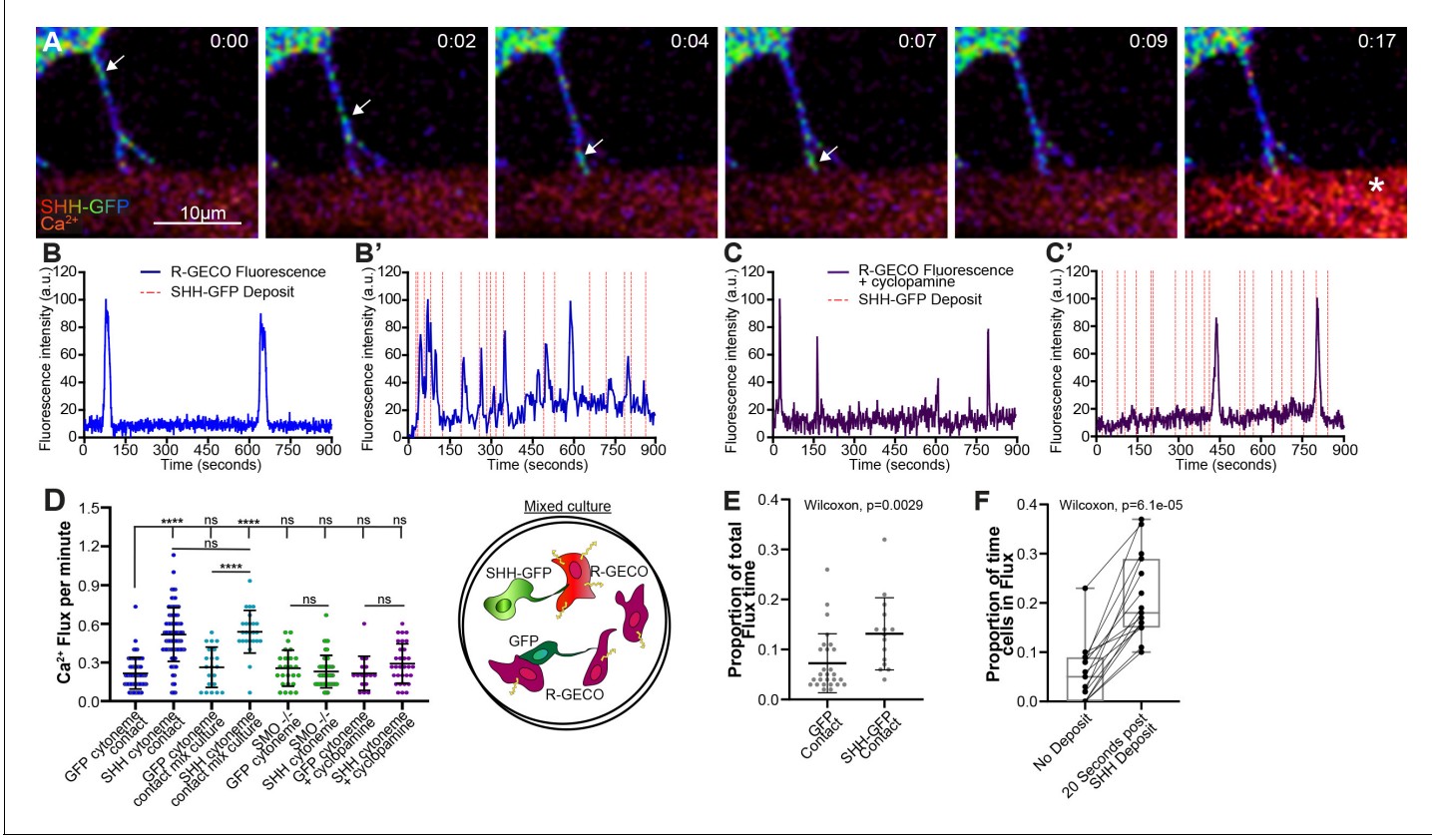

**Figure 2.** Cytoneme-based SHH delivery induces signaling in target cells. (**A**) Time lapse images of an SHH-expressing NIH3T3 cell show SHH-GFP as a fluorescent intensity-spectrum in puncta. Progressive movement of a single punctum traveling down a cytoneme to the tip in contact with a R-GECO sensor cell is indicated by an arrow. R-GECO fluorescent intensity increases ~ 10 s after release of the SHH puncta from the cytoneme (asterisk). Time stamp indicates minutes:seconds. (**B–C'**) R-GECO fluorescent intensity graphs are shown for single cells in contact with a cytoneme from GFP-expressing (**B and C**) or SHH-GFP-expressing (**B' and C'**) cells. Flux activity occurring over 15-min contact periods in the absence (**B,B'**) or presence (**C, C'**) of the inverse SMO agonist cyclopamine is shown. (**D**) Flux rates per minute were calculated for 14–51 individual cells per condition, with visual representation of the mixed culture conditions. (**E**) Total flux time for R-GECO reporter cells is shown (n = 27 for GFP and n = 15 for SHH contact). (**F**) Box plot (min/max whiskers) comparison of individual R-GECO cells (n = 15) receiving SHH-GFP cytoneme deposits measured as proportion of time spent in flux in the absence of an SHH deposit, or within 20 s following deposit. All data are presented as mean ± SD, unless stated otherwise. ns = not significant, ***p<0.001, ****p<0.0001. See also *Figure 2—figure supplement 1*. Source data for (B, B', E and F) can be found in *Figure 2—source data 1*. Source data for (D) is located in *Figure 2—source data 2*.

The online version of this article includes the following source data and figure supplement(s) for figure 2:

**Source data 1.** Normalized fluorescent intensity values of individual R-GECO cells in contact with GFP or SHH-GFP cytonemes.

**Source data 2.** Flux counts per minute of R-GECO cells in contact with cytonemes from specified conditions.

**Figure supplement 1.** Generation and validation of mouse (m)SHH-GFP and SHH-mCherry.

does not contribute to the observed Ca²⁺ response in cells in this assay. Thus, cytonemes can deliver SHH to induce a bona fide SMO activation signal.

## SHH is trafficked inside cytonemes

Our observation that cytonemes appeared to transport distinct puncta of SHH toward target cells (*Figure 2A*) prompted us to investigate the mode by which the morphogen reached the cytoneme tip. HHs are covalently modified by cholesterol at their carboxyl-termini, raising the possibility that they could travel along cytoneme membranes inside cholesterol-rich lipid rafts (*Callejo et al., 2011*; *Creanga et al., 2012*; *Porter et al., 1996*; *Rietveld et al., 1999*). To test for SHH localization to rafts along NIH3T3 cytoneme membranes, we assayed for ligand colocalization with a fluorescently-labeled cholera toxin (CTX) raft marker. Although rafts were evident along the length of cytonemes, they rarely contained SHH, as evidenced by a negative correlation coefficient between CTX and SHH

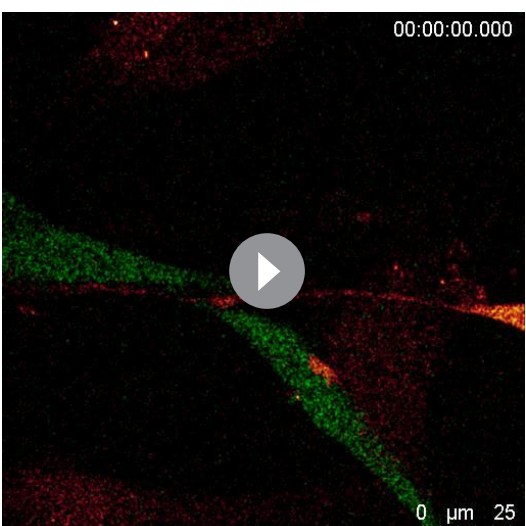

**Video 4.** NIH3T3 R-GECO-positive cells in contact with GFP expressing cells. GFP-expressing NIH3T3 cells (green) are shown in contact with R-GECO reporter cells (orange intensity spectrum) and presented as a maximum intensity projection of 4 Z-sections spanning 3 μm, imaged at 1.7 s/frame over 15 min. Time stamp indicates hr:min:s.
https://elifesciences.org/articles/61432#video4

exosomal markers CD9-mCherry and CD81-mCherry

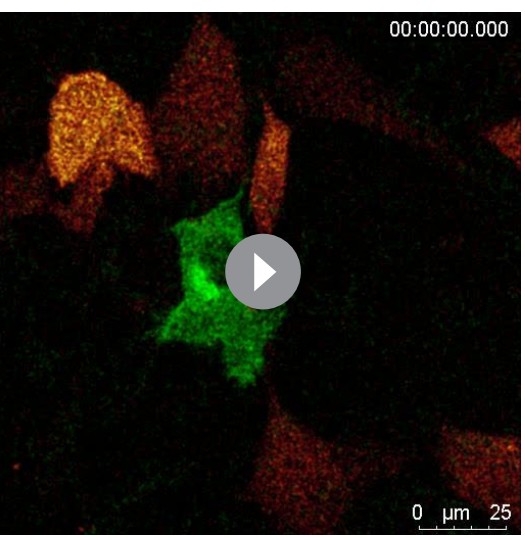

**Video 5.** NIH3T3 R-GECO-positive cells in contact with cytonemes from SHH-GFP expressing cell. Cytonemes from SHH-GFP-expressing cells (green) are shown contacting R-GECO reporter cells (orange intensity spectrum), shown as a maximum intensity projection of 4 Z-sections spanning 2.5 μm, imaged at 1.7 s/frame over 15 min. Time stamp indicates hr:min:s.
https://elifesciences.org/articles/61432#video5

signals (*Figure 3A–B*). Thus, SHH is unlikely to transport along cytoneme membranes in raft-like domains.

We next considered that SHH might load into vesicular structures for transport inside cytonemes because studies in both *Drosophila* and mouse suggest that HH ligands are released from producing cells in exosomes (*Coulter et al., 2018*; *Gradilla et al., 2014*). To investigate this, we tested whether SHH localized inside cytonemes, or to the outside leaflet of cytoneme membranes. Cells expressing SHH-GFP were subjected to extracellular immuno-staining with anti-SHH antibody prior to fixation. GFP fluorescence was used to track total SHH and antibody signal (ex-SHH) was used to monitor the ligand pool exposed to the extracellular environment. SHH-GFP and ex-SHH signals were both evident in puncta on the plasma membrane of SHH-expressing cells (*Figure 3C*, arrows; *Figure 3—figure supplement 1A–D* for antibody controls). Notably, surface aggregates of ex-SHH evident along membranes of the cell body were rarely seen along cytonemes. Conversely, SHH-GFP signal was consistently detected in cytonemes, suggesting SHH is positioned inside cytoneme membranes (*Figure 3C′*). Consistent with this hypothesis, SHH colocalized with the tetraspanin along cytonemes (*Figure 3D,E*), suggesting ligand likely traffics inside cytonemes through a vesicular transport mechanism. Although HH family ligands have been reported to enrich in RAB18 and CD63 containing exosomes in mouse neuronal cells and *Drosophila* tissues, respectively, SHH failed to colocalize with these markers in cytonemes of NIH3T3 cells (*Figure 3—figure supplement 1E–G*; *Coulter et al., 2018*; *Gradilla et al., 2014*). Thus, cell-type-specific vesicular loading may occur.

To evaluate whether SHH trafficked through cytonemes in vesicles, SHH-mCherry-expressing NIH3T3 cells were examined by immuno-electron microscopy using anti-mCherry antibody (*Figure 3F* and *Figure 3—figure supplement 2* for control experiments). Transmission electron microscopy was performed on 70 nm sections of cells and their cytonemes. Cytonemes of SHH-producing cells contained multiple vesicles (*Figure 3F*, arrowheads), many of which were positive for SHH-mCherry (*Figure 3F,F′* arrows, *Figure 3—figure supplement 2A*). Consistent with what was observed with anti-exSHH (*Figure 3C*), clusters observed along the plasma membrane of the cell body were not evident along cytoneme membrane in TEM sections (*Figure 3—figure supplement 2B*).

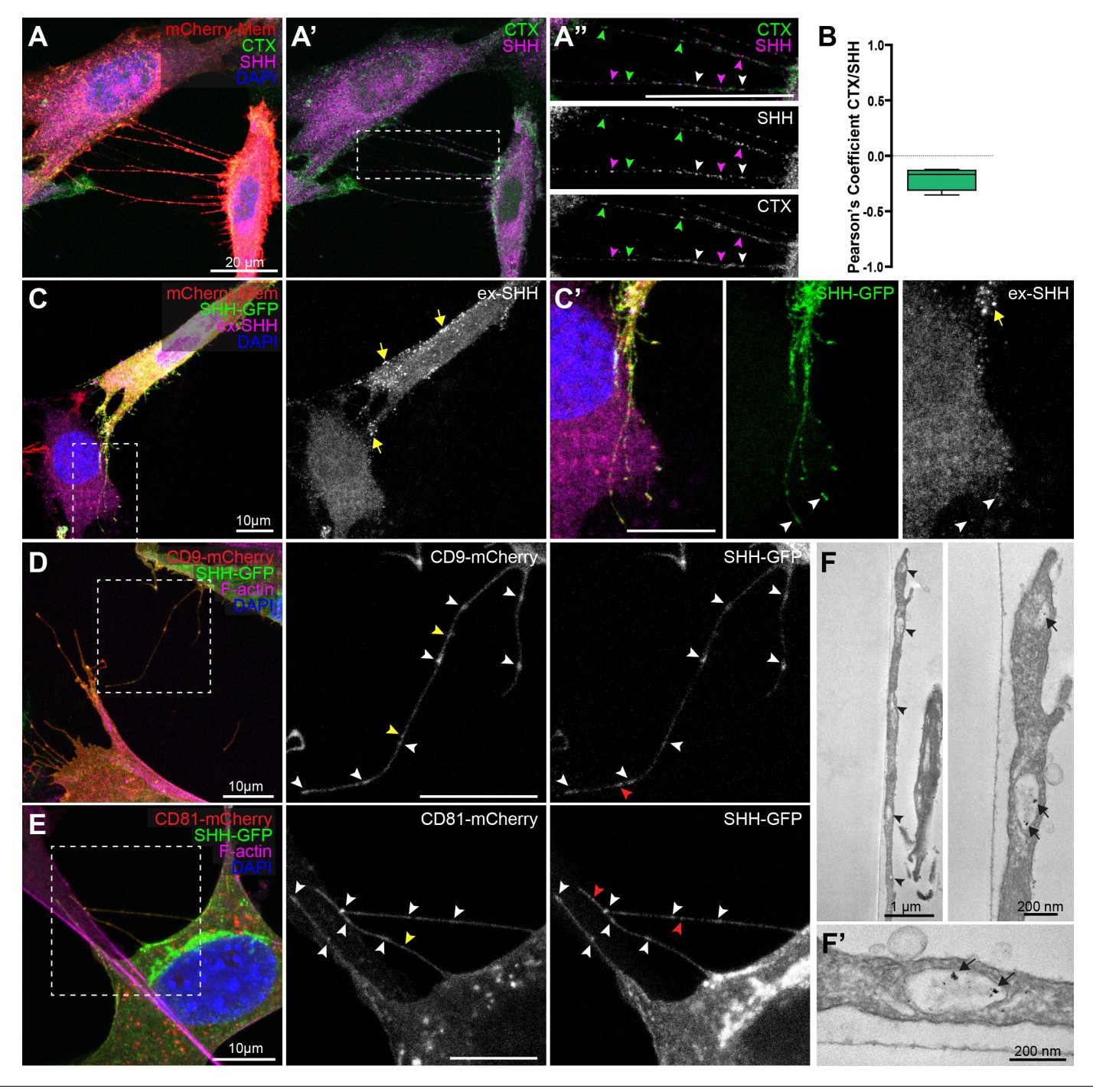

**Figure 3.** SHH is transported inside cytonemes. (A–A'') NIH3T3 cells expressing SHH (magenta) and mCherry-Mem (red), were incubated with lipid raft marker CTX (green). (A') shows CTX and SHH, and (A'') shows channels zoomed in with colored arrowheads indicating SHH or CTX puncta. White arrowheads mark rare colocalization events between SHH and CTX along cytonemes. (B) Box plot with min/max whiskers of Pearson's correlation coefficients values of colocalization between SHH and CTX in cytonemes. (C,C') NIH3T3 cell expressing SHH-GFP and mCherry-Mem is shown. Although SHH-GFP (green) is detected along cytonemes and at tips (C', arrowheads), staining for extra-cellular SHH (ex-SHH, magenta and white) shows signal only on the cell body (C,C', yellow arrows). (D,E) CD9-mCherry (D) and CD81-mCherry (E) colocalize with SHH-GFP in puncta along cytonemes (white arrowheads). CD9/CD81 puncta lacking SHH are indicted by yellow arrowheads. SHH-GFP puncta lacking CD9/CD81 are indicted by red arrowheads. (F,F') Transmission electron microscopy sections of an SHH-mCherry expressing cell cytoneme immunolabeled with anti-mCherry. Cytonemes contain vesicles (F, arrow heads), a subset of which contain SHH-mCherry (F,F' arrows). For all panels, nuclei are marked by DAPI (blue). See also *Figure 3—figure supplements 1* and *2*.

*Figure 3 continued on next page*

*Figure 3 continued*

The online version of this article includes the following figure supplement(s) for figure 3:

**Figure supplement 1.** Control experiments for non-permeabilization immuno-staining of SHH and GFP and colocalization of SHH with lipid rafts, CD63 or Rab18 in cytonemes.

**Figure supplement 2.** Immuno-TEM of SHH-mCherry in NIH3T3 cell cytonemes.

## Myosin 10 promotes cytoneme-based SHH transport

We hypothesized that if SHH undergoes vesicular trafficking inside cytonemes, a molecular motor would likely contribute to its movement. Because MYO10 colocalized with SHH at cytoneme tips (*Figure 1C'*), we tested whether inhibition of MYO10 would impact ligand movement along the specialized filopodia. MYO10-dependent effects on SHH cytoneme mobility were assayed by monitoring fluorescence recovery after photobleaching (FRAP) of the two proteins. Cytonemes of NIH3T3 cells expressing cytoplasmic GFP, mCherry-Mem, MYO10-GFP, or SHH-mCherry were photobleached, and recovery of each protein to cytoneme tips was calculated in the absence or presence of ionomycin, which is proposed to attenuate MYO10 motor activity through raising intracellular $Ca^{2+}$ (*Homma et al., 2001*; *Morgan and Jacob, 1994*). SHH-mCherry and MYO10-GFP cytoneme signals recovered at similar rates in vehicle-treated cells ($\sim0.25 \pm 0.12$ and $0.27 \pm 0.15$ µm/s, respectively), but failed to recover following ionomycin treatment. Conversely, membrane diffusion rates, which were calculated by monitoring mCherry-Mem recovery to cytoneme tips, were not reduced by ionomycin treatment (*Figure 4A–B*, *Figure 4—figure supplement 1*, *Videos 6* and *7*). Recovery of cytoplasmic GFP signal to cytoneme tips was so rapid we were unable to calculate accurate recovery rates in either condition. These results suggest SHH is unlikely to travel along cytonemes through cytoplasmic or membrane diffusion-based mechanisms, and is instead actively transported along the specialized filopodia, potentially by MYO10.

To test whether SHH enrichment in cytonemes required MYO10 function, we generated MYO10-null MEFs from *Myo10^{m1J/m1J}* mutant mice, and assessed SHH cytoneme dynamics in this genetic background (*Heimsath et al., 2017*). SHH was expressed in MYO10 mutant MEFs, and cytoneme to cell body SHH signal intensity ratios were determined (*Figure 4C*). MYO10 mutant MEFs exhibited low SHH cytoneme to cell body signal intensity ratios, indicating inefficient cytoneme enrichment of the morphogen in the absence of MYO10. Enrichment of SHH in cytonemes was restored by co-expression of wild type or pleckstrin homology domain-deficient (ΔPH) MYO10, but not by a MYO10 mutant lacking its cargo binding domains (MYO10-HMM) (*Berg and Cheney, 2002*). Comparable effects on SHH cytoneme enrichment were seen in wild type NIH3T3 cells over-expressing these MYO10-GFP variants (*Figure 4—figure supplement 2A–F*). Although wild type and MYO10ΔPH did not alter the ratio of SHH or SHH-mCherry in cytonemes, MYO10-HMM over-expression reduced SHH cytoneme localization and attenuated SHH-mCherry FRAP to cytoneme tips (*Figure 4D*, *Figure 4—figure supplement 2A–F*). All MYO10 variants showed similar cytoneme FRAP rates, suggesting that failure of SHH to be transported by MYO10-HMM was likely due to compromised cargo binding, and not due to alteration of MYO10-HMM motor activity (*Figure 4—figure supplement 2G*).

MYO10 contributes to the formation and maintenance of filopodia, and SHH increases cytoneme occurrence rates in HEK, NIH3T3, and wild-type MEFs (*Figure 1D,E*, *Figure 1—figure supplement 1F*; *Bohil et al., 2006*). Hence, we next tested for a role for MYO10 in SHH-stimulated cytoneme occurrence in wild-type and *Myo10* mutant cells. In NIH3T3 cells, expression of wild-type MYO10 modestly enhanced the ability of SHH to increase cytoneme occurrence rates over baseline. MYO10-HMM over-expression suppressed occurrence rates (*Figure 4E*), which we speculate resulted from the mutant protein oligomerizing with endogenous MYO10 to disrupt its ability to bind and transport cargo (*Figure 4E* and *Figure 4—figure supplement 2G*). In *Myo10^{-/-}* MEFs, SHH failed to stimulate cytoneme occurrence. Cytoneme occurrence rate increases in the presence of SHH were rescued by reintroduction of either wild type MYO10 or MYO10-ΔPH, but not by MYO10-HMM (*Figure 4E'*), further supporting an essential role for the cargo domain for SHH cytoneme biology. We next tested the ability of GFP-MYO10-HMM expressing cells to deliver an SHH activation signal. Consistent with our hypothesis, co-expression of GFP-MYO10-HMM with SHH in ligand-producing NIH3T3 cells reduced SHH-induced $Ca^{2+}$ flux in co-cultured R-GECO reporter cells to significantly

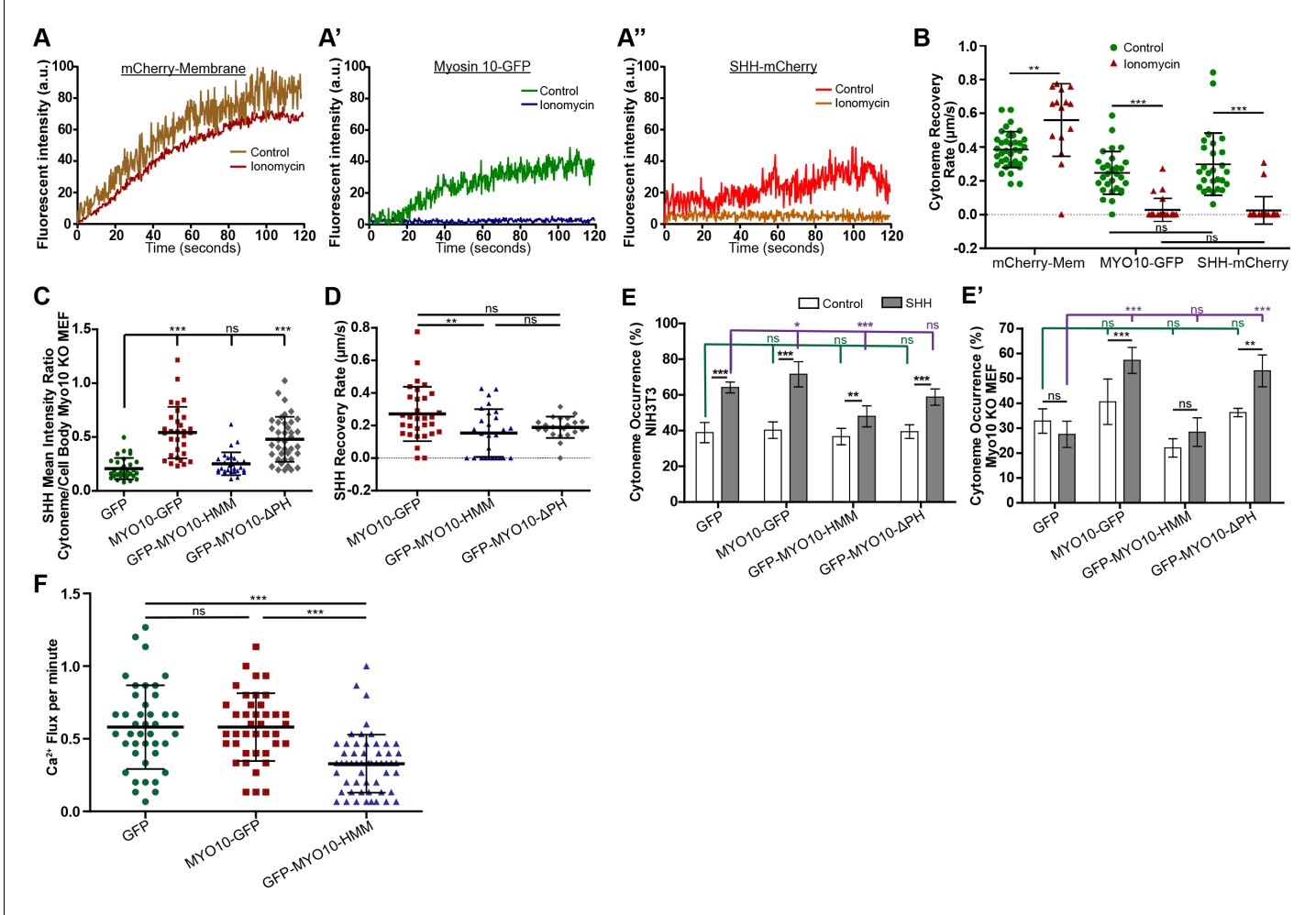

**Figure 4.** Myosin10 facilitates cytoneme-based SHH transport and delivery. (A–A'') NIH3T3 cells expressing the indicated fluorescent proteins were subjected to FRAP. Representative FRAP curves of (A) mCherry-Mem, (A') MYO10-GFP, and (A'') SHH-mCherry to a cytoneme tip in control (DMSO) or ionomycin-treated (2.5 µM) conditions are shown. (B) Scatter plots show the recovery rates of the indicated fluorescent proteins to cytoneme tips in control (DMSO) or ionomycin-treated conditions, calculated from FRAP curves (n = 14–38 cells). (C) Scatter plots of mean cytoneme to cell body SHH fluorescent signal ratios in *Myo10$^{-/-}$* MEFs co-expressing GFP or the indicated MYO10 proteins (n = 28–38). (D) Scatter plots of FRAP calculated recovery rates for SHH-mCherry movement toward cytoneme tips in NIH3T3 cells co-expressing the indicated MYO10 proteins (n = 22–32). (E,E') Cytoneme occurrence rates were calculated for MEM-fixed NIH3T3 and *Myo10$^{-/-}$* MEFs co-expressing mCherry-Mem (control) or SHH plus the indicated MYO10 proteins or GFP control. (F) Ca$^{2+}$ flux rates per minute were determined for R-GECO reporter cells in contact with cytonemes of SHH-producing cells co-expressing the indicated MYO10 proteins (n = 40–55 cells per condition). Data are represented as mean ± SD. See also *Figure 4—figure supplements 1* and *2*.

The online version of this article includes the following figure supplement(s) for figure 4:

**Figure supplement 1.** FRAP of NIH3T3 cell cytonemes treated with DMSO (control) or ionomycin.

**Figure supplement 2.** MYO10 influences SHH transport in cytonemes.

lower rates than those observed upon co-culture with ligand-producing cells co-expressing GFP or wild type MYO10-GFP with SHH (*Figure 4F*). Combined, these results suggest MYO10 is required for the cytoneme-promoting effects of SHH, and also for cargo domain-mediated transport of SHH to cytoneme tips for delivery to target cells.

## Myosin 10 promotes SHH signaling in vivo

*Myo10*-null mice (*Myo10$^{m1J/m1J}$*) are semi-lethal with ~60% of homozygous mutants exhibiting exencephaly and embryonic or perinatal lethality. Surviving animals display white belly spots, with a subset of these animals also exhibiting syndactyly (*Bachg et al., 2019*; *Heimsath et al., 2017*).

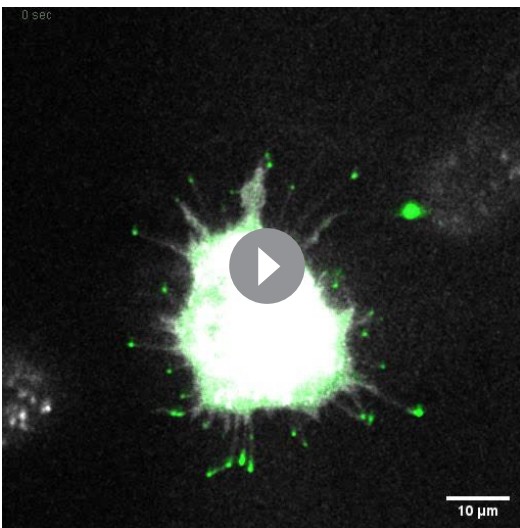

**Video 6.** FRAP of NIH3T3 cytonemes expressing SHH-mCherry and MYO10-GFP treated with DMSO. SHH-mCherry (white) and MYO10-GFP (green) recover to the cytoneme tip within 120 s post photobleaching. Cell was imaged at two frames per second from a single focal plane for 5 s prior to photobleaching, following ~130 s of recovery.

https://elifesciences.org/articles/61432#video6

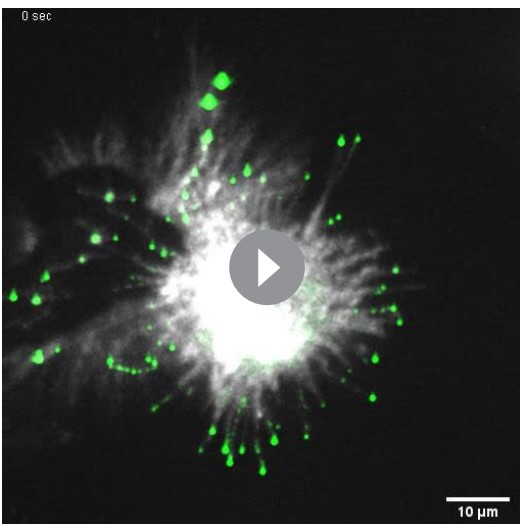

**Video 7.** FRAP of NIH3T3 cytonemes expressing SHH-mCherry and MYO10-GFP treated with ionomycin dissolved in DMSO. SHH-mCherry (white) and MYO10-GFP (green) do not recover to the cytoneme tip post photobleaching. Recovery of SHH-mCherry occurs along the base of some cytonemes but fails to reach the tip. Live images were acquired as described for *Video 6*.

https://elifesciences.org/articles/61432#video7

Exencephaly and syndactyly can be attributed to reduced Bone Morphogenic Protein (BMP) signaling and de-repression, rather than disruption, of SHH signaling (*Nikolopoulou et al., 2017*; *Patterson et al., 2009*). These phenotypes are seemingly inconsistent with our in vitro observations that MYO10 promotes cytoneme stability and transport of SHH (*Figure 4A–F*). Therefore, in an effort to understand the effects of MYO10 loss on SHH signaling in vivo, we analyzed developing neural tubes from $Myo10^{m1J/m1J}$ E9.5 embryos (*Heimsath et al., 2017*). SHH expressed in the notochord signals to the adjacent floor plate of the developing neural tube to induce SHH, which then signals in a ventral to dorsal trajectory to specify neural progenitor domains. Control of these domains is exquisitely sensitive to alteration of SHH signaling, so monitoring their induction allows for robust analysis of gradient function (*Kutejova et al., 2016*; *Placzek and Briscoe, 2018*). In order to track the zone of ventral SHH activity in MYO10 mutant animals, we introduced an *Shh::GFP* allele in which GFP is inserted into the endogenous *Shh* locus such that the mature ligand retains an internal GFP tag adjacent to the carboxyl-terminal cholesterol modification (*Chamberlain et al., 2008*). Examination of neural tubes from $Shh^{GFP/+}$ and $Shh^{GFP/+}$; $Myo10^{m1J/m1J}$ neural tubes revealed altered floor plate induction, as evidenced by reduced SHH::GFP signal in the ventral-most neural tube (*Figure 5A–D*). Consistent with attenuated SHH signaling activity in neural tubes of MYO10-null animals, the expression domain of SHH-induced *Gli1* was compressed in $Myo10^{m1J/m1J}$ embryos with exencephaly (*Figure 5E–G*). This correlated with attenuated SHH-controlled progenitor domain induction, as indicated by reduced *Olig2* expression (*Figure 5K–M*). Moreover, we noted a consistent delay in initiation of notochord regression in MYO10-null animals, further supporting attenuated GLI activity (*Figure 5H–J*; *Park et al., 2000*). These results support a role for MYO10 in SHH morphogen gradient function in vivo.

## SHH colocalizes with DISP and co-receptors in cytonemes

Having identified MYO10 as a new functional player in cytoneme occurrence and SHH transport, we next wanted to determine whether known SHH-binding partners would also impact cytoneme occurrence in mammalian cells. Experiments in *Drosophila* and chick model systems suggest a role for the SHH deployment protein DISP and co-receptors BOC/BOI and

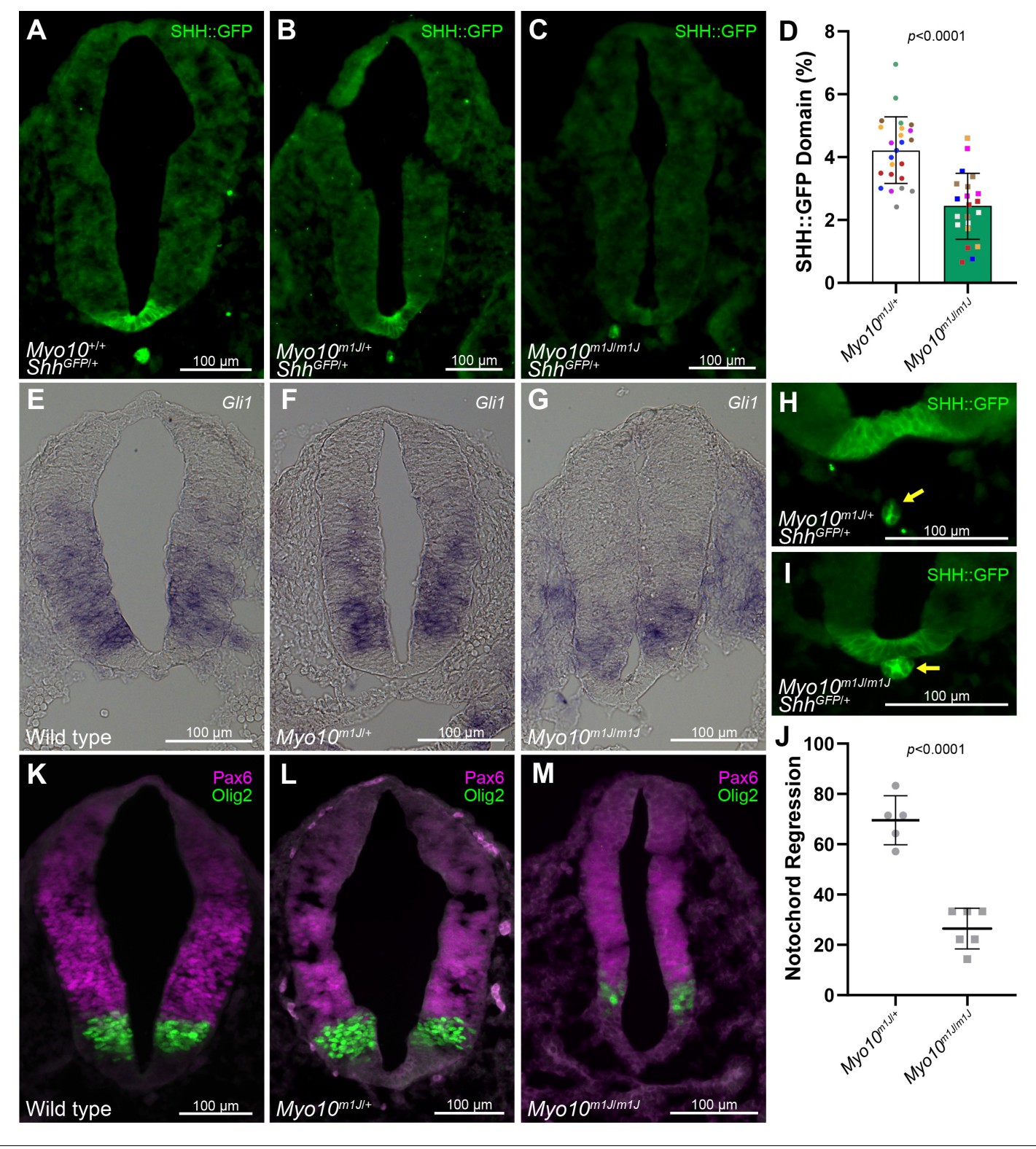

**Figure 5.** MYO10 influences the SHH morphogen gradient. (**A–C**) Representative section of cardiac-level neural tube from E9.5 (**A**) $Shh^{GFP/+}$; $Myo10$ wild type, (**B**) $Shh^{GFP/+}$; heterozygous or (**C**) $Shh^{GFP/+}$; null mice were immuno-stained for GFP. The domain of SHH::GFP in the floorplate was reduced in $Myo10^{m1J/m1J}$ mice. (**D**) Relative SHH::GFP neural tube area in $Myo10^{m1J/+}$ and $Myo10^{m1J/m1J}$ sections. Dots are color-coded for each embryo and represent the individual sections examined. Seven $Myo10^{m1J/+}$ and six $Myo10^{m1J/m1J}$ embryos per condition were analyzed, with 3–4 sections per embryo. (**E–G**) RNA in situ hybridization with $Gli1$ probe. The $Gli1$ expression domain is reduced in $Myo10^{m1J/m1J}$ sections. (**H–I**) Notochord (arrow)

*Figure 5 continued on next page*

Figure 5 continued

regression initiates in (H) $Shh^{GFP/+}$; $Myo10^{m1J/+}$, but it delayed in (I) $Shh^{GFP/+}$; $Myo10^{m1J/m1J}$ embryos. (J) Percentage of sections showing notochord regression of >5 µm from the floorplate was calculated across 5–7 cardiac level section in 5 $Myo10^{m1J/+}$ and 6 $Myo10^{m1J/m1J}$ embryos. Dots represents individual embryos analyzed. (K–M) Wild type, $Myo10^{m1J/+}$ and $Myo10^{m1J/m1J}$ E9.5 neural tubes were immuno-stained for $Olig2$ and $Pax6$. Data are represented as mean ± SD.

CDON/iHOG in cytoneme function (*Bodeen et al., 2017*; *Callejo et al., 2011*; *González-Méndez et al., 2017*; *Gradilla et al., 2014*; *Sanders et al., 2013*). In flies, DISP promotes cytoneme stability of ligand producing cells, and in chick, BOC stabilizes cytonemes of SHH receiving cells (*Bodeen et al., 2017*; *Sanders et al., 2013*). Because iHOG has been reported to stabilize cytonemes and localize to exovesicles from ligand-producing cells in flies (*González-Méndez et al., 2017*; *Gradilla et al., 2014*), we hypothesized that DISP and CDON or BOC might function together to influence cytoneme occurrence and/or function in mouse cells. DISP-HA and GFP-tagged BOC or CDON were co-expressed in NIH3T3 cells in the absence and presence of SHH, and colocalization between the three proteins was assessed. Confocal microscopy revealed that all three proteins localized to cytonemes, but that DISP did not significantly colocalize with either BOC or CDON along cytoneme membranes in the absence of SHH (*Figure 6A,D,F,G*). Co-expression of SHH increased colocalization between BOC and DISP throughout cytoneme membrane and in SHH-positive puncta (*Figure 6A–B', F*). Notably, puncta containing all three proteins were evident in cells abutting SHH-containing cytonemes (*Figure 6B* and zoom in B', arrowheads). As such, DISP and BOC may be released to target cells along with ligand, as has been reported for iHOG in *Drosophila* (*Gradilla et al., 2014*). Consistent with ligand-containing endosomes being internalized by receiving cells, immunoelectron microscopy revealed early and late endosomal structures containing SHH near cytoneme contact points on the signal-receiving cell (*Figure 6C*).

To better understand the associations between cytoneme-localized DISP, BOC, and SHH, stimulated emission depletion (STED) microscopy was used to examine individual SHH puncta within cytonemes. STED showed DISP and SHH consistently positioned adjacent to BOC, suggesting that a trimeric complex may occur in cytonemes in the presence of ligand (*Figure 6H,I*). To test for an interaction between the three proteins, co-immunoprecipitation experiments were performed using lysates from NIH3T3 cells expressing DISP-FLAG and BOC-EGFP in the absence and presence of SHH. BOC-EGFP was captured on anti-FLAG beads in the absence of SHH, suggesting that DISP and BOC can associate (*Figure 6J*, and *Figure 6—figure supplement 1A* for the uncropped blot). Upon ligand expression, SHH incorporated into DISP-FLAG/BOC-EGFP immunocomplexes without significantly altering BOC binding (*Figure 6J*). Thus, SHH is not required for biochemical association between DISP and BOC, but may promote enrichment of the trimeric complex in cytonemes.

BOC and CDON are semi-redundant for co-receptor function in PTCH-SHH binding, but do show differential expression and functionality in temporal and tissue-specific contexts (*Allen et al., 2011*; *Bergeron et al., 2011*; *Cardozo et al., 2014*; *Okada et al., 2006*; *Tenzen et al., 2006*). Likely consistent with context-dependent functionality, co-localization dynamics between DISP and CDON differed from what was observed for DISP and BOC. CDON colocalized with DISP at SHH-positive puncta in cytonemes, albeit to a slightly lesser extent than did BOC (*Figure 6D–G*). However, unlike what was observed for BOC, ligand expression did not increase colocalization between DISP and CDON along the length of cytonemes (*Figure 6F,G*). Biochemical interrogation of DISP-CDON binding by immunoprecipitation analysis revealed that whereas CDON-GFP was captured on FLAG beads by DISP-FLAG in the absence of ligand, its association with DISP-FLAG was reduced upon SHH-DISP binding (*Figure 6K*, *Figure 6—figure supplement 1B* for the uncropped blot). Thus, distinct functional pools of CDON with differential affinity toward DISP ± SHH may exist.

## BOC and CDON promote SHH cytoneme formation

*Drosophila* iHOG/CDON and chick BOC proteins have been reported to localize to cytonemes and influence their behavior in vivo (*Callejo et al., 2011*; *González-Méndez et al., 2017*; *Sanders et al., 2013*). To determine the effects of BOC and CDON on cytonemes of SHH-expressing murine cells, the co-receptors were expressed with SHH in MEFs, and cytoneme occurrence rates were determined. The vertebrate-specific SHH co-receptor GAS1, which has not yet been investigated for a role in cytonemes, was also tested (*Allen et al., 2007*). GAS1-expressing cells showed baseline and

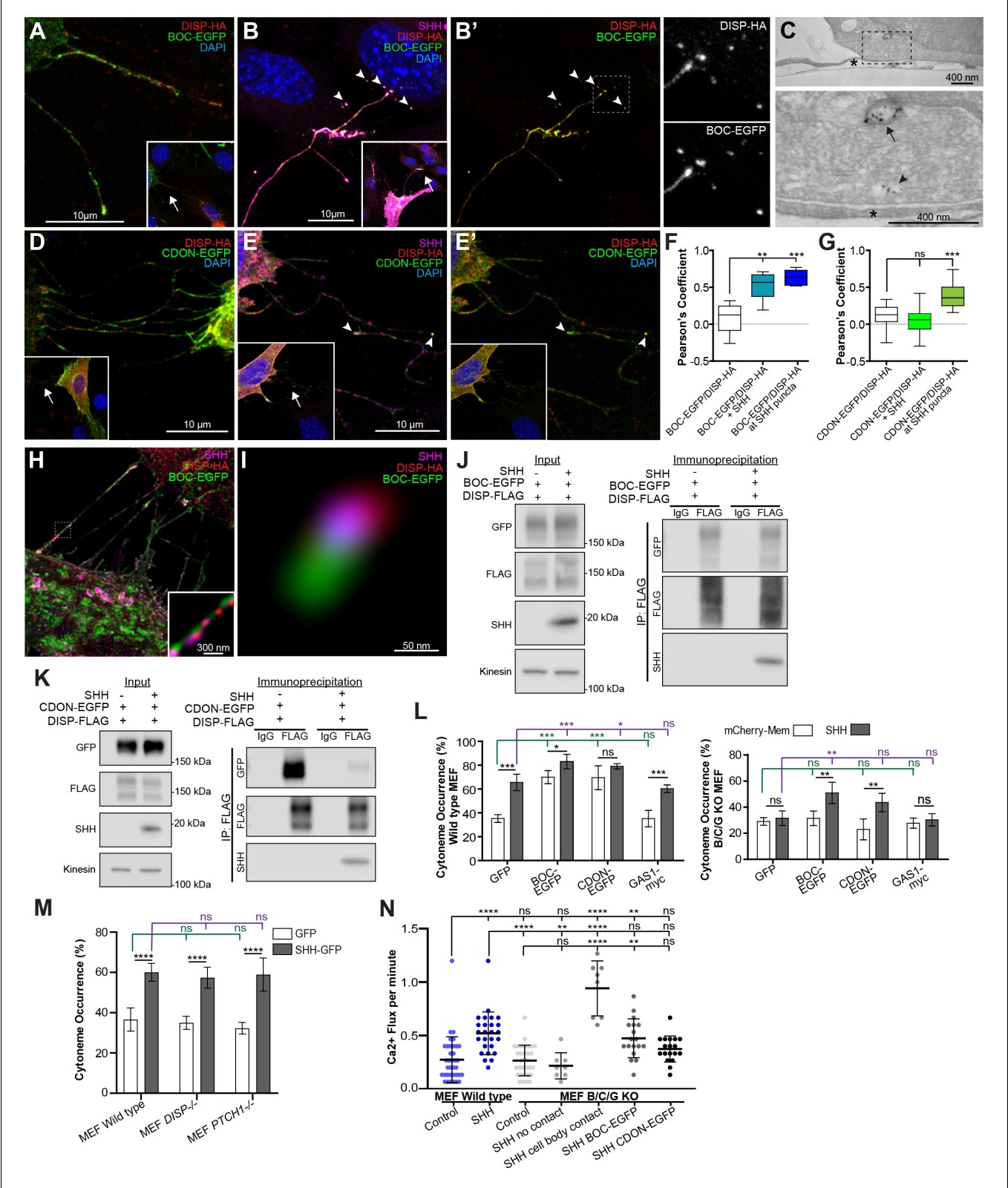

**Figure 6.** DISP, BOC, and CDON ligand complexes influence SHH cytonemes. (**A**) DISP-HA (red) and BOC-EGFP (green) localize to cytonemes of NIH3T3 cells. Inset shows lower magnification, with arrow indicating magnified area. (**B–B'**) SHH (magenta)-expressing NIH3T3 cell cytonemes show co-localization between DISP-HA and BOC-EGFP. Inset shows lower magnification, with arrow indicating magnified area. Puncta visible on signal-receiving cells contain BOC-EGFP and DISP-HA (arrowheads and right). Dashed lines indicate magnified regions, right. (**C**) TEM section of an SHH-mCherry

*Figure 6 continued on next page*

*Figure 6 continued*

expressing cell cytoneme in contact with a receiving cell. SHH is immunolabeled with anti-mCherry and is present in early (arrow head), and late (arrow) endosomal compartments of the receiving cell near the cytoneme contact point (asterisk). (D) DISP-HA and CDON-EGFP (green) are present in cytonemes of NIH3T3 cells. Inset shows lower magnification, with arrow indicating magnified area. (E–E′) DISP-HA, CDON-EGFP, and SHH localize to cytonemes of NIH3T3 cells. Arrowheads identify colocalization. (F–G) Box plots of Pearson's correlation coefficient value (min/max whiskers) measuring colocalization between (F) DISP-HA and BOC-EGFP, or (G) DISP-HA and CDON-EGFP under the indicated conditions. (n=>30 cytonemes per condition). (H–I) Stimulated emission depletion microscopy images of SHH, BOC-EGFP, and DISP-HA in cytonemes (H) and a representative magnified puncta (I). (J–K) DISP-FLAG was immunoprecipitated from lysate from NIH3T3 cells expressing DISP-FLAG and BOC-EGFP (J) or CDON-EGFP (K) in the absence or presence of SHH. Lysate input is shown at left and FLAG immunoprecipitates are shown at right. IgG serves as control. (L) Cytoneme occurrence rates for wild type and *Boc/Cdon/Gas1-/-* (BCG KO) MEFs expressing mCherry-Mem or SHH in the presence of GFP or the indicated SHH co-receptor. (M) Cytoneme occurrence rates for wild type, *Ptch1^{-/-}* and *Disp1^{-/-}* are shown in GFP or SHH-GFP-expressing cells (N). $Ca^{2+}$ flux was determined in R-GECO reporter cells in contact with cytonemes (or cell body) of the indicated ligand-producing cells. BCG KO cells fail to produce cytonemes that contact neighboring cells, so do not promote cytoneme-mediated signal initiation in R-GECO reporter cells. BCG KO cells can induce a response in reporter cells through direct cell body contact. Cytoneme-based signal initiation is fully or partially rescued by BOC or CDON re-expression in SHH-expressing cells. All data are presented as mean ± SD, unless stated otherwise. ns = not significant, *p<0.05, **p<0.01, ***p<0.001, ****p<0.0001. See also *Figure 6—figure supplement 1*.

The online version of this article includes the following figure supplement(s) for figure 6:

**Figure supplement 1.** Western blots in support of *Figure 4J–K*.

SHH-induced cytoneme occurrence rates similar to GFP-expressing control cells, indicating that GAS1 over-expression does not actively promote or stabilize cytonemes in the absence or presence of ligand (*Figure 6L*). Conversely, both BOC and CDON expression elevated baseline cytoneme occurrence rates near to SHH-stimulated levels, and modestly enhanced the ability of SHH to increase occurrence (*Figure 6L*). To determine whether BOC, CDON, GAS1 or a combination of the co-receptors was required for SHH-induced cytoneme occurrence rate increases, SHH was expressed in *Boc^{-/-}*, *Cdon^{-/-}*, *Gas1^{-/-}* triple KO MEFs (BCG KO) (*Allen et al., 2011*), and occurrence rates were quantified in control and co-receptor re-expressed conditions (*Figure 6L*, right panel). Unlike *Ptch^{-/-}* or *Disp1^{-/-}* cells, which showed cytoneme occurrence increases in response to SHH expression, BCG KO cells failed to increase cytoneme occurrence in response to SHH expression (*Figure 6L–M*). BCG KO cells were also compromised in their ability to induce a $Ca^{2+}$ response in co-cultured R-GECO reporter cells in the absence of direct cell body contact, indicating that at least one of the co-receptors is required to facilitate SHH-induced cytoneme occurrence and delivery (*Figure 6L–N*). Consistent with previous reports that PTCH can be activated by SHH that is tethered to neighboring cell membranes (*Caspary et al., 2002*; *Tokhunts et al., 2010*), BCG KO cells were able to induce a $Ca^{2+}$ response in R-GECO reporter cells upon direct cell-cell body contact (*Figure 6N*). GAS1 re-expression did not rescue the ability of SHH to promote cytoneme occurrence, further supporting that GAS1 is not a modulator of SHH cytoneme function. Re-expression of BOC rescued SHH-mediated cytoneme occurrence increases in the triple KO cells, and restored the ability of cells to deliver an activation signal from BCG KO cells stably expressing ligand to co-cultured $Ca^{2+}$ reporter cells (*Figure 6L,N*). CDON re-expression also restored cytoneme occurrence and receiving-cell signal induction, but not as effectively as BOC. Thus, we conclude that either BOC or CDON are required for SHH-induced cytoneme biogenesis or stability, and that BOC is likely the predominant co-receptor functioning in the specialized filopodia.

## Discussion

The formation of a morphogen gradient of sufficient robustness to confer tissue patterning is a complex process that likely involves integration of multiple molecular mechanisms promoting morphogen release and transport. SHH morphogen is unique in that it harbors two essential lipid modifications, including an amino-terminal palmitate and a carboxy-terminal cholesterol, that must be overcome to facilitate release from producing cell membranes for transport through the extracellular milieu (*Pepinsky et al., 1998*; *Porter et al., 1996*). Reported dissemination mechanisms proposed to neutralize the lipid modifications include sheddase-directed release, in which the lipids are cleaved, free diffusion of multimeric SHH in which the multimer configuration buries the lipids, and assisted diffusion, in which extracellular chaperones counteract hydrophobic behavior of the lipids

(*Creanga et al., 2012*; *Ohlig et al., 2012*; *Tukachinsky et al., 2012*; *Zeng et al., 2001*). Proposed deployment mechanisms in which lipid-modified ligand is packaged or transported include exosome-mediated deployment, lipoprotein particle association, and transport through specialized filopodia called cytonemes (*Eugster et al., 2007*; *Gradilla et al., 2014*; *Ramírez-Weber and Kornberg, 1999*). Despite growing bodies of work supporting each of these transport mechanisms, the molecular machinery driving them, how they coordinate activity, and the cell and tissue contexts in which they occur remain unclear. Herein, we investigated cytoneme-based transport of SHH. We focused on molecular components facilitating movement and delivery of SHH ligands through the specialized filopodia for activation of signaling in target cells. High-resolution live imaging microscopy combined with our improved ability to fix cytonemes of cultured cells, allowed us to interrogate the cell biology of SHH cytonemes, identify new components of the cellular machinery contributing to cytoneme-based SHH delivery, and validate contribution of one of the components in vivo.

By developing and validating an assay in which ligand-activated SMO-induced Ca$^{2+}$ release can be monitored in real time, we were able to demonstrate direct, contact-dependent induction of a SHH response within seconds of cytoneme-mediated ligand delivery. Using this assay as a probe, we identified roles for the actin motor MYO10 and the adhesion co-receptors BOC and CDON during cytoneme-based SHH transport and delivery. Consistent with its reported role in filopodial outgrowth (*Bohil et al., 2006*), Myo10$^{-/-}$ MEFs showed an approximate 10% reduction in basal cytoneme occurrence rates compared to control MEFs, and failed to increase occurrence rates following SHH expression. Thus, MYO10 is likely important for cytoneme biogenesis. Our results suggest that the molecular motor also plays a specific role in cytoneme-based transport of SHH that is facilitated through the MYO10 cargo binding domain. This hypothesis is supported by the observations that SHH and MYO10 traffic along cytonemes at similar velocities, and that cells expressing a MYO10 cargo binding mutant fail to enrich SHH in cytonemes, or to induce a pronounced SHH signal response in co-cultured R-GECO reporter cells.

In vivo studies of MYO10 mutant mice reveal ventral neural tube patterning defects in animals with exencephaly that are consistent with alteration of the SHH morphogen gradient. These include a reduced zone of SHH activity at the floorplate, compressed *Gli1* and *Olig2* expression domains, and altered notochord regression. In addition we failed to observe SHH expression in the floor plate of a subset of embryos examined. However, floor plate specification was not consistently lost, indicating SHH signaling from the notochord to the floor plate can occur in the absence of MYO10 function. Homozygous mutants that did not show exencephaly survived to adulthood without developing SHH phenotypes. The failure of MYO10 mutants to consistently exhibit embryonic or adult SHH loss-of-function phenotypes is not unprecedented given the variable phenotypic penetrance observed in *Myo10* knockout animals (*Heimsath et al., 2017*). We speculate that functional compensation by cytoneme-independent mechanisms of SHH distribution likely occur to limit the impact of MYO10 loss on SHH notochord to floor plate signaling and morphogen gradient establishment (reviewed in *Hall et al., 2019*). Further, functional redundancy with other actin-based motors may provide some compensation for MYO10 loss. One candidate is MYO5a, which can also localize to filopodia, bind cargo, and traffic toward filopodial tips (*Kerber and Cheney, 2011*). Nevertheless, the observation that the majority of severely affected *Myo10* null embryos show exencephaly may suggest attenuated cytoneme function in these animals. This is because exencephaly can result from attenuated signaling by BMP, the *Drosophila* ortholog of which has been demonstrated to transport along cytonemes (*Nikolopoulou et al., 2017*; *Patterson et al., 2009*; *Roy et al., 2014*).

The mechanism by which MYO10 transports SHH to filopodial tips has yet to be determined. Given that SHH localized to vesicles inside cytonemes, we anticipate that MYO10-dependent movement of ligand occurs through motor-driven vesicular transport along actin filaments. The ΔPH mutant, which lacks phospholipid-binding capability, was not compromised for SHH transport or cytoneme occurrence. Thus, we hypothesize that MYO10 might connect to SHH-containing vesicles through its cargo domain via an adaptor protein localized to vesicular membranes such as a tetraspanin (*Andreu and Yáñez-Mó, 2014*). Consistent with this possibility, immuno-localization studies revealed co-localization of SHH with CD9 and CD81 tetraspanins in discrete puncta along cytonemes.

In addition to identifying MYO10 as a functional partner in cytoneme-based SHH transport, our studies also revealed a novel role for co-receptors BOC and CDON in signal producing cell

cytonemes. Although the co-receptors have previously been reported to localize to cytonemes (*Gradilla et al., 2014*; *Sanders et al., 2013*), our study is the first to reveal that they can associate with the SHH deployment protein DISP, and act in producing cells to promote cytoneme formation and stability for SHH delivery. Despite functional redundancy between the two proteins (*Allen et al., 2011*; *Zheng et al., 2010*), we found that their associations with DISP-SHH complexes differed. Whereas both BOC and CDON associated with DISP in the absence of SHH, the amount of CDON in association with DISP was reduced upon SHH expression. DISP, CDON and SHH were observed to co-localize to puncta along cytonemes, suggesting the minor fraction still in association with DISP in the presence of ligand may be specific to cytoneme function. However, BOC may be the preferred cytoneme-localized co-receptor because it showed greater colocalization with DISP and facilitated a stronger SHH signal response in signal-receiving cells. Notably, either BOC or CDON was required for SHH to promote cytoneme occurrence because cells lacking all co-receptors failed to initiate cytonemes, or to induce long range signals without re-expression of one of the adhesion co-receptors. Importantly, BOC/CDON/GAS1 mutant cells are not compromised for contact-mediated SHH release because the SHH-expressing BCG KO cells can induce a response when directly abutting a signal-receiving cell. As such, we conclude BOC and CDON contribute to SHH transport from producing cells through promoting cytoneme occurrence and/or stability in the presence of ligand.

Determining the mechanism(s) by which SHH-containing co-receptor complexes promote cytoneme induction or stability is beyond the scope of the current study. However, the reported ability of BOC to activate the cytoskeletal regulator JNK during neuronal differentiation (*Vuong et al., 2017*), suggests that BOC or CDON might connect SHH with actin remodelers for cytoneme movement. Thus, our results suggest the exciting possibility that SHH may 'reverse-signal' in an autocrine fashion to promote its own transport. We anticipate that this reverse signal is SMO/GLI independent, and instead, occurs through BOC or CDON co-receptor in complex with SHH. Our observation that SHH expression can increase cytoneme occurrence rates in both *Ptch*$^{-/-}$ and *Disp1*$^{-/-}$ fibroblasts indicates that trimeric co-receptor complexes are not required for cytoneme initiation or stabilization in either signal sending or receiving cells. Thus, PTCH or DISP association with cytoneme-localized co-receptors likely confers specificity for ligand release or canonical signal induction without directly influencing cytoneme behavior. In *Drosophila*, DISP is required for HH to promote cytoneme stability, suggesting functionality of cytoneme-localized DISP may differ between fly and vertebrate systems (*Bodeen et al., 2017*). We do not know how DISP function evolved between the systems, but speculate that the additional SHH-binding proteins such as GAS1 and SCUBE2, which are present in vertebrates and lacking in flies, may account for the discrepancy (*Allen et al., 2007*; *Creanga et al., 2012*).

Future studies will be required to determine how SHH is loaded into cytoneme vesicles along with deployment complex components, and how ligand is transferred from signal producing cytonemes to target cells. The observed accumulation of exSHH puncta along the cell body combined with the paucity of exSHH signal along cytonemes may hint at how ligand enters the specialized filopodia. Studies in *Drosophila* indicate that HH ligands are initially directed to apical membranes where they are re-internalized by DISP prior to release for long range signaling (*Callejo et al., 2011*; *D'Angelo et al., 2015*; reviewed in *Hall et al., 2019*). Thus, the exSHH signal could represent pre-packaged protein that is poised for vesicular loading with BOC/CDON by DISP. It is possible that SHH may fail to load into or be released from specialized filopodia without deployment complex activity because cytoneme membranes may have a composition unique from that of bulk plasma membrane. Filopodia are documented to enrich for select phosphatidylinositol species (*Jacquemet et al., 2019*), which may alter the overall membrane composition to prevent diffusion of SHH onto cytoneme membranes without assistance. Due to improved methods for preserving and imaging cytonemes of cultured cells, and for monitoring responses in signal-receiving cells, we are now poised to address these provocative questions. The proven utility of cultured cells for analyzing cytoneme biology reveals that in vitro systems can function as tractable models for interrogating morphogen transport. Furthermore, cultured cells may also allow for investigation of how cytonemes synergize with other morphogen dispersion processes to ensure gradient robustness during tissue development.

## Materials and methods

### Immunofluorescence and imaging

Cell fixation and staining were performed using MEM-fixation protocols (*Hall and Ogden, 2018*). The following antibodies and dilutions were used: rabbit anti-SHH (H-160) (1:100; Santa Cruz), mouse anti-GFP (4B10) (1:500; CST), rat anti-HA (1:250; Roche), mouse anti-CD63 (E-12) (1:100; Santa Cruz), rabbit anti-Myc-Tag (2272) (1:400; CST). Secondary antibodies (Jackson ImmunoResearch and Invitrogen) were used at a 1:1000 dilution. For additional information please refer to Appendix 1—key resources table. Lipid raft staining was performed using Cholera Toxin Subunit B (Recombinant) (CTX), Alexa Fluor 488 Conjugate (Invitrogen). CTX was dissolved in chilled PBS to a final concentration of 1.0 mg/mL. CTX was incubated with cells for 20 min at 4°C to prevent endocytosis. Cells were rinsed three times in chilled PBS prior to MEM-fixation. Extracellular staining was performed by diluting antibodies in 4°C PBS supplemented with 5% normal goat serum. Antibody solutions were then incubated for 30 min on live cells on ice to prevent endocytosis. Cells were rinsed three times in chilled PBS prior to MEM-fixation. Microscopy images were taken with a TCS SP8 STED 3X confocal microscope (Leica) for fixed and live cell imaging.

### Fluorescence recovery after photobleaching (FRAP)

FRAP assays were carried out on a Bruker Opterra swept field confocal microscope, equipped with an enclosure box at 5% $CO_2$ and imaging and objective heater at 37°C. NIH3T3 cells were imaged in phenol red-free standard growth media. For assays involving ionomycin, standard growth media was replaced immediately prior to imaging with phenol red- and serum-free media supplemented with 0.083% DMSO (control), or DMSO with 2.5 µM ionomycin (#9995, CST). Image acquisition was performed with 60x/1.4NA/Oil objective lens (CFI Plan Apo Lambda) with a 30 µm pinhole array and 70 µm width slit. Fluorescence was recorded with a 5 s baseline followed by a complete photobleaching of a cytoneme with 488 nm and 561 nm lasers. Fluorescence recovery was recorded for 120 s with 100 ms exposure per channel with frames taken every 277 ms. Fluorescence recovery of individual regions of interest (ROI) along the cytoneme to its tip were normalized with pre-FRAP equal to 100% and post-FRAP equal to 0%. FRAP curves were corrected for any loss of fluorescence during acquisition (*Fritzsche and Charras, 2015*).

### Generation of cell lines and culture

Cells were cultured at 37°C in 5% $CO_2$. NIH3T3 (CRL-1658), HEK293T (CRL-11268), and LightII (JHU-68) cells were obtained from ATCC. HEK (*Bosc 23*) ponasterone A inducible SHH cells were obtained from D. Robbins (*Goetz et al., 2006*). *Boc/Cdon/Gas1*$^{-/-}$ MEFs were obtained from *Allen et al., 2011*. *Myo10*$^{-/-}$ MEFs were generated from mice obtained from and cryo-recovered by The Jackson Laboratory (stock number 024583, B6.Cg-Myo10$^{m1J/GrsrJ}$).

*Smo*$^{-/-}$ Flp-In-3T3 cells were generated using CRISPR-Cas9 technology. Briefly, 400,000 Flp-In-3T3 cells were transiently co-transfected with precomplexed ribonuclear proteins (RNPs) consisting of 100 pmol of chemically modified sgRNA (mSmo.sg–NA - 5'- CAGCUACAUCGCAGCCUUCG -3', Synthego), 33 pmol of spCas9 protein (St. Jude Protein Production Core), and 200 ng of pMaxGFP (Lonza). The transfection was performed via nucleofection (Lonza, 4D-Nucleofector X-unit) using solution SG and program EN158 in a small (20 µl) cuvette according to the manufacturer's recommended protocol. Five days post-nucleofection, cells were single cell sorted for GFP+ (transfected) cells by FACs and clonally expanded. Clones were screened and verified for the desired out-of-frame indel modifications via targeted deep sequencing using gene specific primers with partial Illumina adapter overhangs (mSmo.F – 5'- ttccttccccgtgtcagaacgaggt -3' and mSmo.R – 5'- gcggccatgcagtgaagtgagggtc -3', overhangs not shown) as previously described (*Sentmanat et al., 2018*). In brief, clonal cell pellets were lysed and used to generate gene specific amplicons with partial Illumina adapters in PCR#1. Amplicons were indexed in PCR#2 and pooled with other targeted amplicons for other loci to create sequence diversity. Additionally, 10% PhiX Sequencing Control V3 (Illumina) was added to the pooled amplicon library prior to running the sample on an MiSeq Sequencer System (Illumina) to generate paired 2 × 250 bp reads. Samples were demultiplexed using the index sequences, fastq files were generated, and NGS analysis was performed using CRIS.py (*Connelly and Pruett-Miller, 2019*).

MEFs were generated as previously described (*Jozefczuk et al., 2012*). Briefly, pregnant dams were harvested at E12.5–13.5 and embryos were dissected in 1X PBS, then decapitated and internal organs removed. The remaining tissue was rinsed in 1X PBS, then finely minced into pieces in a dish containing Trypsin-EDTA (0.25%) (Gibco). The dish was placed at 37°C in an incubator for 15 min, then an additional 2 mL of Trypsin-EDTA was added, tissue was vigorously pipetted, then placed back in the incubator at 37°C for an additional 10 min. The solution was transferred to a 15 mL conical tube and contents were allowed to settle for 2 min. Supernatant was removed, and then centrifuged for 5 min at 200 x g. The cell pellet was resuspended in MEF media (see below) and plated in a 60 mm plate and left overnight. Each line was then SV40-transformed and single cell selection was performed by serial dilution in a 96-well plate. MEF lines were derived from five different *Myo10* mutant embryos and five wild type littermates.

Cells were maintained in DMEM (Life Technologies) supplemented with 10% bovine calf serum (Fisher Scientific) and 1% Penicillin Streptomycin solution (Gibco). HEK293T cells utilized 10% heat-inactivated fetal bovine serum (Corning). Cell lines were routinely validated by functional assay and western blot as appropriate and tested monthly for mycoplasma contamination by MycoAlert (Lonza). Transfection of plasmid DNA was performed with Lipofectamine 3000 and P3000 reagent (Thermo Fisher Scientific), according to manufacturer's instructions. When required, the final amount of DNA used for transfection was kept constant by the addition of control vector DNA. All cells were harvested 36 hr after transient DNA plasmid transfection for subsequent assays. SHH inducible cells were incubated in HEK media with ponasterone A at the indicated concentrations for 16 hr prior to analysis. Incubation of SMO modulators was performed with 100 nm SAG or 200 nm vismodegib (LC laboratories) for 16 hr prior to analysis.

## In vivo analysis

Wild-type, *Shh::gfp* (JAX # 008466), and *Myo10^{m1J/m1J}* (JAX # 024583) embryos in the C57BL/6 background were harvested and processed for immunohistochemistry at E9.5. Pregnant dams were harvested, uterine horns removed, and embryos were dissected in 1X PBS, then rinsed three times. Embryos were fixed overnight at 4°C in 2% PFA. The following day, embryos were rinsed three times in 1X PBS and moved to 30% sucrose to cryo-protect. The following day, embryos were frozen in O.C.T. Compound (Tissue-Tek) on dry ice. Embryos were sectioned transverse at 10 µm thickness on a Leica Microm CM1950 cryo-stat. Sections were briefly dried, then washed in 1X TBST, then blocked with 2% BSA, 1% goat serum, 0.1% Triton-X-100 in 1X PBS. Antibodies were diluted in blocking buffer and incubated overnight on sections at room temperature. The following antibodies and dilutions were used: chicken anti-GFP (1:500; Aves), mouse anti-PAX6 (1:25; DSHB), and rabbit anti-OLIG2 (1:300; Millipore). Primary antibody was removed, sections were washed with 1X TBST three times, then incubated for 3 hr in secondary antibodies (Invitrogen) used at a 1:500 dilution. Sections were washed three times in 1X TBST, then rinsed with tap water, and cover slips were applied with ProLong Diamond mounting media. Sections were imaged on a Leica DMi8 widefield microscope and processed using LAS X. SHH::GFP neural tube domains were calculated as the mean area per section of the neural tube taken at the cardiac level (n = 3–4 sections per mouse). A minimum of five embryos per genotype were analyzed. For in situ hybridization E9.5 embryos were harvested and fixed for 4 hr at 4°C in 4% PFA. In situ hybridization experiments were performed as described previously (*Abler et al., 2011*) with the following modifications. Frozen embryos were sectioned transverse at 15 µm thickness and mounted on charged glass slides. The sense and antisense digoxigenin-labeled RNA probes were made using a DIG RNA labeling kit and following the manufacturer's instructions (Roche), sense probes were used as negative controls and no positive signal was observed. Three embryos per genotype were analyzed. All *Myo10^{m1J/m1J}* embryos analyzed displayed overt exencephaly.

## Ca$^{2+}$ flux assay

Approximately $0.4 \times 10^6$ NIH3T3 cells were seeded into individual wells of 6-well plates one day prior to transfection. A total of 2 µg plasmid DNA was transfected into individual wells. pCMV-R-GECO1 for 'receiving' sensor cells, and the appropriate construct combination (e.g. SHH-mCherry, GFP, MYO10-GFP, etc...) for the 'producing' cells. Six hours after transfection, cells were trypsinized and replated into eight well, polystyrene chambers on a 1.5 borosilicate coverglass, 0.7 cm$^2$/well

(Nunc Lab-Tek II). Prior to cell addition, chamber wells were divided in half with #0 (0.08–0.13 mm) thick coverslips (Electron Microscopy Sciences) cut to size, with vacuum grease (Dow Corning) added along the lateral edges to retain a liquid-tight barrier. Receiving and producing cells were seeded onto opposite sides of the barrier and allowed to recover overnight. The following day the barrier was removed 3 hr prior to imaging allowing sufficient time for cells to migrate and cytonemes to extend between producing and receiving cells. In GFP and SHH-GFP coculture experiments, cells were seeded into chambered wells at ~40% cell density (20% R-GECO, 10% GFP and 10% SHH-GFP) and allowed to recover for a minimum of 5 hr before imaging. Media was removed prior to imaging, and cells were gently washed in PBS. Immediately prior to any imaging event, media was replaced to remove secreted SHH. In experiments where SMO activation was inhibited, 10 μM cyclopamine (LC Laboratories) was added to R-GECO cells 16 hr prior to imaging.

Live imaging was performed at 37°C, 5% $CO_2$ with resonant scanning for 15 min per area over the entire cytoneme/s depth (~4–6 μm), with Z-steps of ~0.6–1.0 μm. Maximum intensity projections were generated for subsequent analysis of the time-lapses.

## SHH deposits and $Ca^{2+}$ flux quantification

Time-lapses of cells were analyzed using LAS X (Leica). SHH-GFP deposits onto R-GECO sensor cells were recognized if a SHH puncta was detected traversing a cytoneme from a producing cell body to accumulate at the tip, where in a successive frame fluorescence was absent and did not undergo retrograde movement. Puncta were identified by a fluorescent intensity signal >50% than background cytoneme fluorescence by single line scan along a cytoneme. R-GECO fluorescent intensity histograms of individual cells were normalized for each cell with minimum fluorescence equal to 0 and maximum to 100. $Ca^{2+}$ flux occurrence was quantified as a relative peak in R-GECO fluorescence within a ~20 s window with a minimum peak fluorescence of 50. Maintained fluorescence over 20 s was considered a single flux. Total flux time was determined using a threshold to define an increased flux (i.e. a peak) using the R-GECO cells in contact with GFP-control samples. The threshold was determined to be a flux value of greater than 50 which lies between the overall 90th and 95th percentiles of the control samples. Next, the proportion of time (seconds) when flux values were greater than 50 was calculated for each control and SHH case sample. The proportion of time when flux values were greater than 50 was compared between cases and controls using the Wilcoxon rank sum test. For SHH cases only, the proportion of time when flux values were greater than 50 was compared by the occurrence of a SHH deposit (yes vs. no) using the Wilcoxon signed rank test. Twenty seconds was considered a biologically relevant time frame in which a deposit and a subsequent increased flux should occur (*Adachi et al., 2019*; *Tewson et al., 2012*). Statistical analyses were conducted using SAS software version 9.4 (SAS Institute, Cary, NC) and R version 3.6.0 (R Foundation for Statistical Computing, Vienna, Austria). A two-sided significance level of $p<0.05$ was considered statistically significant. R-GECO cells in contact with cytonemes that did not exhibit a single flux during the time-lapse were excluded from analysis.

## Image processing, measurements, and statistical analysis

### Image processing

Following image acquisition, images were processed using LAS X (Leica), and Photoshop 2019 (Adobe), and figures were made using Illustrator 2019 (Adobe). Stimulated emission depletion images underwent deconvolution using automated sampling with Huygens Professional software (Scientific Volume Imaging). Videos were processed with ImageJ and Imaris (Bitplane). Images are presented as maximum intensity projections (MIP) of the z-stack acquisition spanning the cell, unless stated otherwise. Images shown represent a standard cell or tissue section from a minimum of three biological replicates unless specifically stated otherwise.

### Cytoneme metrics

For quantification we defined cytonemes in cultured cells as cellular projections approximately <200 nm in diameter, with a minimum length of 10 μm. Any cellular protrusions that originated from the basal surface of the cell and maintained continuous contact with the coverslip were excluded from analysis (*Hall and Ogden, 2018*). For live imaging, cytonemes were identified as motile protrusions, capable of elongation with the exception if a cytoneme was in contact with a nearby cell body or

other cells' cytonemes. For occurrence rate counts, a minimum of 100 cells per condition were counted, performed on triplicate coverslips with a minimum of two biological replicates.

## Calculation of diffusion/recovery rates of proteins to cytoneme tips
Protein recovery rates to the cytoneme tip were derived by average velocity,

$$\bar{v} = \frac{1}{\tau 1/2 - t0} \int_{t0}^{\tau 1/2} v(t)dt$$

based upon the following conditions. (1) A cytoneme diameter is ~100 nm, below the diffraction limited resolution of the confocal microscope in which the FRAP data was acquired. Therefore, all data may be reduced to a single plane (a 1-dimensional line). (2) Photobleaching of the entire cytoneme allows for a single vector recovery from the cell body, as such 2D diffusion coefficients calculations are not required. FRAP data was analyzed by Igor Pro 8 (WaveMetrics) to calculate halftime recovery ($\tau_{1/2}$) of 2–3 ROIs along an individual cytoneme. $\tau_{1/2}$ is dependent upon the distance from the cell body, allowing for instantaneous velocity calculation.

$$v = \frac{distance\ from\ cell\ body\ (um)}{\tau 1/2\ (s)}$$

Values were then averaged per cytoneme. Sixteen to 39 individual cytonemes were analyzed for each condition.

## Colocalization analysis
Image analysis was performed using CellProfiler (*McQuin et al., 2018*). An image analysis pipeline was constructed to mask and isolate cytonemes. Output images were then run through a secondary pipeline measuring Pearson's correlation coefficient between pixels of different fluorophores to calculate the relative colocalization of the proteins of interest within a cytoneme. A third pipeline was used for determining protein colocalization of two proteins at SHH puncta in cytonemes. This pipeline segmented SHH pixels as a reference point within cytonemes to measure Pearson's correlation coefficient between pixels of the other fluorophores of interest in contact with SHH. A minimum of 30 cytonemes were analyzed per condition.

## Statistical analyses
All analyses were performed using GraphPad Prism. One-way analysis of variance was performed for multiple comparisons, with Tukey's multiple comparison as a posttest. Significant differences between two conditions were determined by two-tailed Student's t tests. All quantified data are presented as mean ± SD, with $p < 0.05$ considered statistically significant. Significance depicted as *$p < 0.05$, **$p < 0.01$, ***$p < 0.001$, ****$p < 0.0001$, ns = not significant.

## Plasmid constructs
The following plasmids were used in this study: pCDNA (control vector) (Clontech), pCDNA3-EGFP (Addgene Plasmid #13031), pCMV-mCherry-Mem (Addgene Plasmid #55779), pcDNA-Wnt3A (Addgene Plasmid #35908), pIRES-Jag1-HA (Addgene Plasmid #17336), pCMV-R-GECO1 (Addgene Plasmid #32444), pCMV-mCherry-CD9 (Addgene Plasmid #55013), pCMV-mCherry-CD81 (Addgene Plasmid #55012), pEGFP-CD63-C2 (Addgene Plasmid #62964), pCMV-EGFP-Rab18 (Addgene Plasmid #49550), pEGFP-C1-hMyoX (Addgene Plasmid #47608), pCMV6-hGAS1-Myc-DDK (Origene Cat: RC224804), pCMV3-mFGF2-N-GFPSpark (SinoBiological Cat: MG50037-ANG), pCMV3-mBMP2-C-GFPSpark (SinoBiological Cat: MG51115-ACG), pCMV-mCherry2, pCDNA3-mSHH-FL, pCDNA3-mSHH-N, pCDNA3-mSHH-FL-EGFP, pCDNA3-mSHH-FL-mCherry2, pCDNA3-V5-Disp-HA, pCS2-hBOC-EGFP and pCS2-hCDON-EGFP (a gift from A. Salic), pEGFP-C2-bMyo10-HMM (*Berg and Cheney, 2002*) and pEGFP-C2-bMyo10-Δ3PH, a modified version of pEGFP-bMyo10 where the 3 PH domains were removed via deletion of aa 1168–1491.

For the generation of fluorescently tagged SHH, GFP or mCherry2 was introduced into the SHH protein immediately 3' to amino acid Gly198 with the addition of 10 amino acids (Alanine188 - AENSVAAKSG - Glycine197) downstream of GFP or mCherry2 including the intein cleavage-cholesterol attachment site, similar to what was previously descried (*Chamberlain et al., 2008*). Briefly, a

Bgl2 site was introduced into SHH-FL after Gly198 by Quikchange (Agilent) using primers (forward 5′ GTGGCGGCCAAATCCGGCGGCAGATCTGGCTGTTTCCCGGGGATCCGCC and reverse 5′ ggcggatcccgggaaacagccagatctgccgccggatttggccgccac). The 10 amino acid duplication 3′ to GFP or mCherry2 on SHH protein was introduced using Infusion (Clontech) using primers (forward 5′TCCGGCGGCAGATCTGCAGAGAACTCCGTGGCGGCCAAATCCGGCGGCTGTTTCCCGGGA and reverse 5′ tcccgggaaacagccgccggatttggccgccacggagttctctgcagatctgccgccgga). GFP or mCherry2 with Bgl2 sites was generated by Phusion PCR (NEB) with the following primers (forward 5′ GAATTCAGATCTATGGTGAGCAAGGGCGAG and reverse 5′ gaattcagatctcttgtacagctcgtccatg) or (forward 5′ GAATTCAGATCTATGGTGAGCAAGGGCGAGGAG and reverse 5′ gaattcagatctcttg-tacagctcgtccatgccg) using pEGFP (Clontech) or pCMV-mCherry2 (Clontech) as the DNA template, respectively.

## Immunoblotting and immunoprecipitation

For western blotting, cells were washed twice in PBS, harvested in 1% NP-40 Lysis Buffer (50 mM Tris-HCl, pH 8.0, 150 mM NaCl, 1% NP-40, 0.1% SDS, 1X Protease Inhibitor Cocktail and 0.5 mM DTT) and incubated for 30 min at 4°C. Extracts were cleared by centrifugation at 14,000 x g at 4°C for 45 min and analyzed. The supernatant was removed, and protein concentrations were deter-mined by bicinchoninic acid (BCA) assay (Pierce). Equal amounts of total protein from each sample were analyzed by SDS-PAGE on Criterion gels (Bio-Rad). SDS-PAGE samples were transferred onto Protran Nitrocellulose (GE) or Immobilon-P PVDF (Millipore) using Tris/Glycine/SDS Buffer (Bio-Rad) at 100V for one hour at 22°C. Membranes were blocked with 5% milk and 0.1% Tween-20 in Tris-buffered saline (TBS) for 1 hr at room temperature. Membranes were immunoblotted for 1 hr at 22°C using the following antibodies: rat anti-HA (1:3000; Roche), mouse anti-V5 (1:5000; Life Technolo-gies), rabbit anti-SHH (H-160) (1:1000; Santa Cruz), rabbit anti-GFP (1:8000, Rockland), rabbit anti-Kinesin (anti-Kif5B, 1:5000; Abcam), mouse anti-α-Tubulin (DM1A) (1:5000, CST), followed by three 5 min washes in secondary milk (primary milk diluted to 25% with TBS). Corresponding HRP-conju-gated secondary antibodies (Jackson Immuno) were incubated for 1 hr at RT at a 1:5000 concentra-tion. Blots were developed using an Odyssey Fc (Li-Cor) with ECL Prime (GE).

For immunoprecipitation assays, proteins of interest were expressed in NIH3T3 cells. Cell lysates were prepared ~48 hr post-transfection using a 0.5% NP-40 Lysis Buffer (30 mM Tris-HCl, pH 7.4, 75 mM NaCl, 0.5% NP-40, 5% glycerol, 2 mM MgCl2, 4 mM KCl, 1 mM EDTA, and 1X Protease Inhibi-tor Cocktail) and incubated for 30 min at 4°C with two units per mL of Benzonase Nuclease to degrade DNA from protein samples. Extracts were cleared by centrifugation at 14,000 x g at 4°C for 30 min, supernatant was collected, and protein concentration was determined by BCA assay (Pierce). Equal total protein amounts for each sample were used in co-immunoprecipitation assays and ana-lyzed by SDS-PAGE on Criterion gels (Bio-Rad). Co-immunoprecipitation assays were performed as described (Stewart et al., 2018) with the following modifications. Samples were pre-cleared with A/G Plus Agarose for 30 min with gentle rotation. Samples were centrifuged at 1000 x g for 1 min and set up in new tubes with either anti-Mouse IgG1 control or EZview Red Anti-Flag Affinity Gel (Sigma) (to immunoprecipitate Flag epitope-tagged proteins) for three hours at 4°C with gentle rotation. Samples were then centrifuged at 1000 x g for 1 min and supernatant was removed. Beads were washed 3x for 5 min each with 0.5% NP-40 Lysis Buffer with gentle rotation at room temperature. Proteins were eluted from agarose beads with 1X SDS sample buffer (2% SDS, 4% v/v Glycerol, 40 mM Tris-HCl, pH 6.8, 0.1% Bromophenol blue) by incubating them at room temperature for 5 min. Samples were centrifuged at 2000 x g for 2 min and the eluent was transferred to a new tube. Immu-noprecipitates were analyzed by western blot using the following antibodies: rabbit anti-GFP (1:8000; Rockland), rabbit anti-Flag (DDDDK) (1:3000, Abcam), rabbit anti-Shh (1:2000; SCBT), and rabbit anti-Kif5B, (1:5000; Abcam).

## Transcriptional reporter assay

For co-culture *Gli*-reporter assays, HEK293T cells were seeded at a density of $1 \times 10^6$ cells per 60 mm plate. The following day, pCDNA3-GFP (2 μg), pCDNA3-SHH-FL-GFP-10aa linker (4 μg), pCMV-mCherry2 (2 μg) and pCDNA3-SHH-FL-mCherry2-10aa linker (4 μg) were transfected into HEK293T cells. In a six-well plate Light II reporter cells were seeded at a density of $0.5 \times 10^6$ cells per well in DMEM-10% FBS complete growth media and grown overnight at 37°C, 5% CO2. The following day,

transfected HEK293T cells underwent trypsinization and were seeded into the Light II wells at a density of $0.5 \times 10^6$ cells per well. Cells were allowed to recover for 4 hr at 37°C, 5% $CO_2$. Media was removed from cells, washed twice with PBS and once with DMEM serum-free complete media (phenol red free). DMEM serum-free media was added back to each well and allowed to incubate for 2 hr. Washing was carried out over 6 hr repeating the above wash steps. After 6 hr, 3 mL of DMEM Serum-free Complete Media was added to each well and the cells were incubated for ~36 hr. Reporter assays were carried out according to Dual Luciferase Reporter Assay Kit instructions (Promega). Experiments were repeated three times in triplicate.

## Electron microscopy

NIH3T3 cells expressing SHH-mCherry were seeded at 60% confluency into eight well, Permanox slide, 0.8 cm²/well (Nunc Lab-Tek II) and, for pre-embedding immunolabeling, were fixed in a 0.5% glutaraldehyde with 4% PFA fixative in 0.1 M phosphate buffer. Prior to labeling with primary antibody, samples were washed with buffer and excess aldehyde groups neutralized with glycine. Samples were blocked with 1% BSA in 10 mM PBS (BSA/PBS) then a blocking solution matched to the species of the secondary antibody (Aurion, Wageningen, The Netherlands) in PBS. Samples were incubated with chicken anti-mCherry (1:1000, Abcam) diluted in BSA/PBS overnight at 4°C. Following primary antibody incubation, samples were washed in BSA/PBS then incubated with a streptavidin conjugated secondary antibody. Samples were rinsed with PBS and incubated with biotinylated nanogold (Nanoprobes) then washed with PBS and fixed in 1% glutaraldehyde (Electron Microscopy Sciences (EMS)). Following fixation, samples were successively rinsed in distilled water and 0.2 M citrate buffer then incubated with HQ Silver Enhancement reagent (Nanoprobes) prepared per manufacturer instructions. Enhancement reaction was halted by rinsing with distilled water. Samples were contrasted successively with 1% osmium tetroxide (EMS) and 1% uranyl acetate (EMS) in water with water washes between contrasting steps. Samples were then dehydrated in an ascending series of alcohols, infiltrated with EmBed 812 (EMS) and polymerized at 80°C overnight. Samples were sectioned on a Leica ultramicrotome (Wetzlar, Austria) at 70 nm and examined in a Tecnai G² F20-TWIN transmission electron microscope. Images were recorded using an AMT side mount camera system. Unless specified, all chemical and reagents were from Sigma.

## Acknowledgements

We thank members of the Ogden lab for thoughtful discussion during the course of this work and for comments on the manuscript. We thank Ben Allen, Adrian Salic and Robert Krauss for BOC, CDON and GAS1 expression vectors and knockout cell lines, Xin Sun for the plasmid encoding the *Gli1* in situ probe and David Robbins for the SHH inducible cell line. *Funding.* This work was supported by National Institute of General Medical Sciences grants R35GM122546 (SKO), R01GM134531 (REC) and by ALSAC of St. Jude Children's Research Hospital. The CTI is supported by the Cancer Center Support Grant, NCI P30 CA021765. The content is solely the responsibility of the authors and may not represent the official views of the National Institutes of Health.

## Additional information

### Funding

| Funder | Grant reference number | Author |
| --- | --- | --- |
| National Institute of General Medical Sciences | R35GM122546 | Stacey K Ogden |
| National Institute of General Medical Sciences | R01GM134531 | Richard E Cheney |
| National Cancer Institute | P30 CA021765 | Camenzind G Robinson |
| American Lebanese Syrian Associated Charities | | |

The funders had no role in study design, data collection and interpretation, or the decision to submit the work for publication.

## Author contributions

Eric T Hall, Miriam E Dillard, Data curation, Formal analysis, Validation, Investigation, Visualization, Methodology, Writing - review and editing; Daniel P Stewart, Ben Wagner, Data curation, Validation, Investigation, Visualization, Methodology; Yan Zhang, Data curation, Formal analysis, Investigation, Methodology; Rachel M Levine, Resources, Formal analysis, Validation; Shondra M Pruett-Miller, Resources, Validation, Writing - review and editing; April Sykes, Formal analysis; Jamshid Temirov, Software, Formal analysis, Visualization, Methodology; Richard E Cheney, Resources, Investigation, Writing - review and editing; Motomi Mori, Formal analysis, Supervision, Validation, Methodology, Writing - review and editing; Camenzind G Robinson, Data curation, Formal analysis, Supervision, Visualization, Methodology, Project administration, Writing - review and editing; Stacey K Ogden, Conceptualization, Resources, Formal analysis, Supervision, Funding acquisition, Writing - original draft, Project administration, Writing - review and editing

## Author ORCIDs

Eric T Hall (iD) https://orcid.org/0000-0001-9656-1942
Shondra M Pruett-Miller (iD) https://orcid.org/0000-0002-3793-585X
Camenzind G Robinson (iD) http://orcid.org/0000-0002-7277-692X
Stacey K Ogden (iD) https://orcid.org/0000-0001-8991-3065

## Ethics

Animal experimentation: The study was performed per recommendations in the Guide for the Care and Use of Laboratory Animals of the National Institutes of Health. All animals were handled according to the approved institutional animal care and use committee protocol number 608-100616-10/19 of St. Jude Children's Research Hospital. All effort was made to minimize suffering.

## Decision letter and Author response

Decision letter https://doi.org/10.7554/eLife.61432.sa1
Author response https://doi.org/10.7554/eLife.61432.sa2

# Additional files

### Supplementary files

• Transparent reporting form

### Data availability

All data generated or analyzed during the study are included in the manuscript and supporting files. Source files are provided for Figure 2.

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

# Appendix 1

**Appendix 1—key resources table**

| Reagent type (species) or resource | Designation | Source or reference | Identifiers | Additional information |
|---|---|---|---|---|
| genetic reagent (*Mus musculus*) | B6.129X1(Cg)-Shhtm6Amc/J | The Jackson Laboratory (JAX) | #008466, RRID: IMSR_JAX:008466 | |
| genetic reagent (*Mus musculus*) | B6.Cg-Myo10m1J/GrsrJ | JAX | #024583, RRID: IMSR_JAX:024583 | |
| genetic reagent (*Mus musculus*) | C57BL/6J | JAX | #000664, RRID: IMSR_JAX:000664 | |
| cell line (*Mus musculus*) | NIH3T3 | ATCC | CRL-1658, RRID: CVCL_0594 | |
| cell line (*Homo sapiens*) | HEK293T | ATCC | CRL-11268, RRID: CVCL_1926 | |
| cell line (*Mus musculus*) | Dispatched KO MEFs | Ma et al., Cell 2002 111(1): 63-75 | | |
| cell line (*Mus musculus*) | Light II | ATCC | JHU-68, RRID: CVCL_2721 | |
| cell line (*Homo sapiens*) | HEK (Bosc 23) SHH inducible | *Goetz et al., 2006* | | |
| cell line (*Mus musculus*) | Boc/Cdon/Gas1 KO MEFs | *Allen et al., 2011* | | |
| cell line (*Mus musculus*) | Myo10 KO MEFs | This study | | Derived from RRID:IMSR_JAX: 024583 |
| cell line (*Mus musculus*) | Ptch1 KO MEFs | Kim et al, Sci. Signaling 2015 8 (379) | | A gift from Phil Beachy |
| cell line (*Mus musculus*) | Smo KO 3T3 | This study | | originated from CRL-1658 |
| cell line (*Mus musculus*) | MEF wt | This study | | C57BL/6 MEF cells |
| antibody | anti-HA (Rat monoclonal) | Roche | 11867423001, RRID: AB_390918 | 1:250 (IF) 1:3000 (WB) |
| antibody | anti-Shh (H-160) (Rabbit polyclonal) | Santa Cruz Biotechnolgy | sc-9024, RRID:AB_ 2239216 | 1:100 (IF) 1:2000 (WB) |
| antibody | anti-GFP (4B10) (Mouse monoclonal) | Cell Signaling Technology (CST) | #2955, RRID:AB_ 1196614 | 1:500 (IF) |
| antibody | anti-CD63 (E-12) (Mouse monoclonal) | Santa Cruz Biotechnolgy | sc-365604, RRID:AB_ 10847220 | 1:100 (IF) |
| antibody | anti-Myc-Tag (Rabbit polyclonal) | CST | #2272, RRID:AB_ 10692100 | 1:400 (IF) |
| antibody | anti-GFP (Chicken polyclonal) | Aves | GFP-1010, RRID:AB_ 2307313 | 1:500 (IF) |
| antibody | anti-PAX6 (Mouse monoclonal) | DSHB | PAX6, RRID:AB_ 528427 | 1:25 (IF) |
| antibody | anti-OLIG2 (Rabbit polyclonal) | Millipore | AB9610 | 1:300 (IF) |
| antibody | anti-GFP (Rabbit polyclonal) | Rockland | 600-401-215, RRID: AB_828167 | 1:8000 (WB) |
| antibody | anti-Kif5b (Rabbit monoclonal) | Abcam | ab167429, RRID:AB_ 2715530 | 1:5000 (WB) |

*Continued on next page*

*Appendix 1—key resources table continued*

| Reagent type (species) or resource | Designation | Source or reference | Identifiers | Additional information |
|---|---|---|---|---|
| antibody | anti-α-Tubulin (DM1A) (Mouse monoclonal) | CST | #3873 | 1:5000 (WB) |
| antibody | anti-Flag (DDDDK) (Rabbit polyclonal) | Abcam | ab1162, RRID:AB_298215 | 1:3000 (WB) |
| antibody | anti-mCherry (Chicken polyclonal) | Abcam | ab205402, RRID:AB_2722769 | 1:1000 (IF) |
| antibody | AlexaFluor 488 | Life technologies | A11029 (Mouse); A11034 (Rabbit); A11006 (Rat) | 1:1000 |
| antibody | AlexaFluor 555 | Life technologies | A21424 (Mouse); A21429 (Rabbit); A21434 (Rat) | 1:1000 |
| antibody | AlexaFluor 633 | Life technologies | A21236 (Mouse); A21245 (Rabbit); A21247 (Rat) | 1:1000 |
| antibody | AlexaFluor 488 F(ab')2 | CST | #4408 (Mouse); #4412 (Rabbit) | 1:1000 |
| antibody | AlexaFluor 555 F(ab')2 | CST | #4409 (Mouse); #4413 (Rabbit) | 1:1000 |
| antibody | AlexaFluor 594 F(ab')2 | Jackson Immuno | #712-586-153, RRID: AB_2340691 | 1:1000 |
| antibody | anti-ATTO 655 STED (Goat anti-Rabbit polyclonal) | Active motif | #15049 | 1:1000 |
| antibody | Peroxidase AffiniPure (Donkey anti-Mouse polyclonal) | Jackson Immuno | 715-035-151, RRID: AB_2340771 | 1:5000 |
| antibody | Peroxidase AffiniPure (Donkey anti-Rabbit polyclonal) | Jackson Immuno | 711-035-152, RRID: AB_10015282 | 1:5000 |
| antibody | Peroxidase AffiniPure (Goat anti-Rat polyclonal) | Jackson Immuno | 112-035-175, RRID: AB_2338140 | 1:5000 |
| antibody | anti-Flag M2 Affinity gel (Mouse monoclonal) | Millipore | F2426 | 25ul slurry for IP |
| transfected construct (Aequorea victoria) | pCDNA3-EGFP | Addgene | RRID:Addgene_13031 | Doug Golenbock |
| transfected construct (Discosoma sp.) | pCMV-mCherry-Mem | Addgene | RRID:Addgene_55779 | Catherine Berlot |
| transfected construct (*Homo sapiens*) | pcDNA-Wnt3A | Addgene | RRID:Addgene_35908 | Marian Waterman |
| transfected construct (R. norvegicus ) | pIRES-Jag1-HA | Addgene | RRID:Addgene_17336 | Joan Conaway, Ronald Conaway |
| transfected construct (synthetic construct) | pCMV-R-GECO1 | Addgene | RRID:Addgene_32444 | Robert Campbell |
| transfected construct (*Homo sapiens*) | pCMV-mCherry-CD9 | Addgene | RRID:Addgene_55013 | Michael Davidson |

*Continued on next page*

*Appendix 1—key resources table continued*

| Reagent type (species) or resource | Designation | Source or reference | Identifiers | Additional information |
|---|---|---|---|---|
| transfected construct (*Homo sapiens*) | pCMV-mCherry-CD81 | Addgene | RRID:Addgene_55012 | Michael Davidson |
| transfected construct (*Homo sapiens*) | pEGFP-CD63-C2 | Addgene | RRID:Addgene_62964 | Paul Luzio |
| transfected construct (*Homo sapiens*) | pCMV-EGFP-Rab18 | Addgene | RRID:Addgene_49550 | Marci Scidmore |
| transfected construct (*Homo sapiens*) | pEGFP-C1-hMyoX | Addgene | RRID:Addgene_47608 | Emanuel Strehler |
| transfected construct (*Homo sapiens*) | pCMV6-hGAS1-Myc-DDK | Origene | RC224804 | |
| transfected construct (*Mus musculus*) | pCMV3-mFGF2-N-GFPSpark | SinoBiological | MG50037-ANG | |
| transfected construct (*Mus musculus*) | pCMV3-mBMP2-C-GFPSpark | SinoBiological | MG51115-ACG | |
| transfected construct | pCMV-mCherry2 | Addgene | RRID:Addgene_54517 | Michael Davidson |
| transfected construct (*Mus musculus*) | pCDNA3-mSHH-FL | *Stewart et al., 2018* | | |
| transfected construct (*Mus musculus*) | pCDNA3-mSHH-N | *Stewart et al., 2018* | | |
| transfected construct (*Mus musculus*) | pCDNA3-mSHH-FL-EGFP | This paper | | |
| transfected construct (*Mus musculus*) | pCDNA3-mSHH-FL-mCherry2 | This paper | | |
| transfected construct (*Mus musculus*) | pCDNA3-V5-Disp-HA | *Stewart et al., 2018* | | |
| transfected construct (*Homo sapiens*) | pCS2-hBOC-EGFP | Wierbowski et al., 2020 | | Gift from Adrian Salic |
| transfected construct (*Homo sapiens*) | pCS2-hCDON-EGFP | Wierbowski et al., 2020 | | Gift from Adrian Salic |
| transfected construct (*Bos taurus*) | pEGFP-C2-bMyo10-HMM | *Berg and Cheney, 2002* | | |
| transfected construct (*Bos taurus*) | pEGFP-C2-bMyo10-Δ3PH | This paper | | |
| Recombinant DNA reagent | mGli1 in situ probes | Hui et al., 1994 | | A gift from Xin Sun |
| software, algorithm | Photoshop 2020 | Adobe | RRID:SCR_014199 | for making figures |

*Continued on next page*

*Appendix 1—key resources table continued*

| Reagent type (species) or resource | Designation | Source or reference | Identifiers | Additional information |
|---|---|---|---|---|
| software, algorithm | Illustrator 2020 | Adobe | RRID:SCR_010279 | for making figures |
| software, algorithm | LAS X | Leica | RRID:SCR_013673 | image processing |
| software, algorithm | Prism 8 | GraphPad | RRID:SCR_002798 | statistical analyses and graph generation |
| software, algorithm | Huygens Professional software | Scientific Volume Imaging | RRID:SCR_014237 | decovolution |
| software, algorithm | Igor Pro 8 | Wavemetrics | RRID:SCR_000325 | FRAP analyses |
| software, algorithm | Imaris | Bitplane | RRID:SCR_007370 | video processing |
| software, algorithm | ImageJ | National Institutes of Health | RRID:SCR_002285 | image analysis and video processing |
| software, algorithm | SAS software | SAS Institute | RRID:SCR_008567 | Statistical analyses |
| software, algorithm | R version 3.6.0 | R Foundation for Statistical Computing | | Statistical analyses |
| software, algorithm | CellProfiler | *McQuin et al., 2018* | RRID:SCR_007358 | image analysis |
| software, algorithm | CRIS.py | *Connelly and Pruett-Miller, 2019* | | NGS analysis |
| commercial assay or kit | Dual Luciferase Reporter Assay Kit | Promega | PRE1960 | |
| commercial assay or kit | ECL Prime Western Blotting Detection Reagent | Fisher Scientific | RPN2232 | |
| commercial assay or kit | Quickchange II XL Kit | Agilent | 200522 | |
| commercial assay or kit | Lipofectamine 3000 | ThermoFisher Scientific | L3000008 | |
| commercial assay or kit | MycoAlert Mycoplasma Detection Kit | Lonza | LT07-118 | |
| chemical compound, drug | Cholera Toxin Subunit B (Recombinant) (CTX) Alexa Fluor 488 Conjugate | Invitrogen | C34775 | |
| chemical compound, drug | ionomycin | CST | #9995 | |
| chemical compound, drug | Ponasterone A | Sigma Aldrich | P3490-1MG | |
| chemical compound, drug | Vismodegib | LC Laboratories | NC1633974 | |
| chemical compound, drug | SAG | Selleck Chemical Co | S7779-2MG | |
| chemical compound, drug | cyclopamine | LC Laboratories | C-8700 | |

