## [Decision Letter]

**Acceptance summary:**

This study addresses a poorly understood question, namely, the role of cytonemes in Sonic Hedgehog (SHH) morphogen delivery and signaling in mammalian cells. Using live cell imaging and a sensitive Ca reporter of SHH-induced SMO activation, the authors provide strong support for the existence of similar cytonemes as those reported in *Drosophila* and show that these structures are involved in intercellular transmission of SHH signals. Their studies also provide insights into the protein composition of the mammalian cytonemes.

**Decision letter after peer review:**

Thank you for submitting your article "Cytoneme Delivery of Sonic Hedgehog Requires Myosin 10 and a Dispatched-BOC/CDON Co-receptor Complex" for consideration by *eLife*. Your article has been reviewed by three peer reviewers, one of whom is a member of our Board of Reviewing Editors, and the evaluation has been overseen by Marianne Bronner as the Senior Editor. The reviewers have opted to remain anonymous.

The reviewers have discussed the reviews with one another and the Reviewing Editor has drafted this decision to help you prepare a revised submission.

Summary:

This study addresses a poorly understood question, namely, the role of cytonemes in Sonic Hedgehog (SHH) morphogen delivery and signaling in mammalian cells. Using live cell imaging and a sensitive Ca reporter of SHH-induced SMO activation, the authors provide strong support for the existence of similar cytonemes as those reported in *Drosophila* and show that these structures are involved in intercellular transmission of SHH signals. Their studies also provide some insights into the protein composition of the mammalian cytonemes. These findings are of general interest, thus appropriate for the broad readership of *eLife*.

Essential revisions:

While the reviewers find the data to be overall convincing and supporting the author's conclusions, they have raised some points that the authors should address in a revision.

1) The authors show that SHH has a dramatic stimulatory effect on cytoneme occurrence. Is this effect due to SHH signaling through the canonical Patched-Smoothened pathway? This could be tested using a synthetic *Smo* antagonist such as vismodegib (not cyclopamine). Along the same lines, does the Smoothened activator, SAG1, cause a similar increase in cytoneme number?

2) The neural tube phenotype of MYO10^-/-^ embryos is very subtle (if it exists at all). A good comparison is the phenotype of *Disp^-/-^* embryos (see Figure 8 in PMID: 12421714), where the SHH ligand fails to exit its site of synthesis in the notochord. *In Disp^-/-^* embryos the floor plate (marked by FoxA2) fails to be specified and so fails to secrete SHH. In contrast, in MYO10^-/-^ embryos, floor plate specification seems fine, showing normal SHH transport. Also the embryo sections seem to be from slightly different stages, making it difficult to determine if the slight change in the *nkx.2* and *olig2* domains is due to a decrease in SHH responsive or just due to differences in timing. For example, embryos in B appear to be for an earlier stage (narrower tube) compared to embryos in A. It is critical in these embryos to show (1) antibody stain for SHH to show that its spread is compromised and (2) in situ hybridization for the direct target gene *Gli1* to show reduction in Hh target gene induction. Unlike the progenitor domains, these are the most direct readouts of SHH transport in the tube and the downstream effect on Shh signal reception.

---

## [Author Response]

Essential revisions:While the reviewers find the data to be overall convincing and supporting the author's conclusions, they have raised some points that the authors should address in a revision.1) The authors show that SHH has a dramatic stimulatory effect on cytoneme occurrence. Is this effect due to SHH signaling through the canonical Patched-Smoothened pathway? This could be tested using a synthetic Smo antagonist such as vismodegib (not cyclopamine). Along the same lines, does the Smoothened activator, SAG1, cause a similar increase in cytoneme number?

We quantified cytoneme occurrence rates in GFP (control) or SHH-expressing NIH3T3 cells following treatment with SAG and Vismodegib. These results are presented in Figure 1—figure supplement 1G. SAG treatment did not significantly increase cytoneme occurrence rates in GFP (white bars) or SHH (gray bars) expressing cells. We tested whether SMO loss would impact cytoneme occurrence by generating SMO-null NIH3T3 cells using CRISPR/Cas9 technology. *Smo^-/-^* NIH3T3 cells showed increased cytoneme occurrence in response to SHH expression, albeit to a slightly lesser extent than what was observed in wild type NIH3T3 cells. Thus, SMO signaling is not required for cytoneme induction, but may enhance cytoneme occurrence rates under SHH-stimulated conditions. As requested, we also examined the effect of Vismodegib, and found that cytoneme dynamics were altered in both the absence and presence of SHH. Vismodegib-treated cells showed elevated baseline occurrence rates and blunted SHH-stimulated cytoneme occurrence, resulting in a modest ligand-induced occurrence rate change in drug treated cells. Nevertheless, given the ability of SHH to increase cytoneme occurrence in a statistically significant manner in *Smo^-/-^* cells, we are confident that canonical PTCH-SMO signaling is not required for the SHH effects.

2) The neural tube phenotype of MYO10^-/-^ embryos is very subtle (if it exists at all). A good comparison is the phenotype of Disp^-/-^ embryos (see Figure 8 in PMID: 12421714), where the SHH ligand fails to exit its site of synthesis in the notochord. In Disp^-/-^ embryos the floor plate (marked by FoxA2) fails to be specified and so fails to secrete SHH. In contrast, in MYO10^-/-^ embryos, floor plate specification seems fine, showing normal SHH transport. Also the embryo sections seem to be from slightly different stages, making it difficult to determine if the slight change in the nkx.2 and olig2 domains is due to a decrease in SHH responsive or just due to differences in timing. For example, embryos in B appear to be for an earlier stage (narrower tube) compared to embryos in A. It is critical in these embryos to show (1) antibody stain for SHH to show that its spread is compromised and (2) in situ hybridization for the direct target gene Gli1 to show reduction in Hh target gene induction. Unlike the progenitor domains, these are the most direct readouts of SHH transport in the tube and the downstream effect on Shh signal reception.

We agree with the reviewers that shifts in progenitor domains in the E10.5 neural tubes shown in the initial submission were difficult to appreciate. To address this concern, we performed several additional crosses to examine stage-matched embryos at E9.5, rather than E10.5. All sections shown were taken from equivalent cardiac level regions of the embryos.

As suggested, we examined the SHH activity domain surrounding the floor plate and performed in situ hybridization to track *Gli1* expression. We also include new neural tube sections showing shifted *Olig2* and *Pax6*-marked progenitor domains. As discussed in the text, the severity of the *Myo10^m1J/m1J^* phenotype is variable, likely due to compensation by other factors. So, we examined only *Myo10^m1J/m1J^* embryos showing overt exencephaly to control against phenotypic variability.

We attempted to stain for SHH using available antibodies, but could not obtain quality results due to pronounced signal to noise issues. So, to address the question about SHH spread in the neural tube, we obtained the *SHH::GFP* allele originally published by the McMahon lab (B6.129X1 (Cg)-Shh^tm6Amc^/J), which tags endogenously-expressed bioactive SHH with GFP. We crossed the *SHH::GFP* allele into the *Myo10^m1J/+^* background, and analyzed the SHH-GFP signal in E9.5 *Myo10^m1J/+^* and *Myo10^m1J/m1J^* embryos. Analysis of SHH^GFP/+^; *Myo10 ^m1J/m1J^* embryos with exencephaly revealed the domain of SHH-GFP signal to be compressed as compared to neural tubes of *Shh^GFP/+^; Myo ^m1J/+^* controls (Figure 5A-C and quantified in D using 3-5 cardiac region sections from 7 individual animals). In addition, *Myo10 ^m1J/m1J^* embryos consistently showed defective initiation of notochord regression, indicative of reduced GLI activity (Park et al., 2000). We speculate near-normal floor plate induction in *Myo10 ^m1J/m1J^* embryos may result from sustained close proximity of the non-regressed notochord to the neural tube, which may allow for SHH to be directly transferred/delivered through cytoneme-independent mechanisms. We briefly discuss this possibility in the Discussion.

As requested, we performed in situ hybridization experiments for the direct SHH target *Gli1* in wild type, *Myo10 ^m1J/+^* and *Myo10 ^m1J/m1J^* E9.5 embryos. *Myo10 ^m1J/m1J^* animals showed clear reduction in *Gli1* expression. E9.5 *Myo10 ^m1J/m1J^* embryos also showed compressed *Olig2* progenitor domains, consistent with reduced SHH target gene induction.

We think the inclusion of these additional in vivo experiments strengthens our conclusion that *Myo10* contributes to SHH deployment in vivo, and thank the reviewers for these helpful suggestions.